



Title: Nocturnal aerosol optical depth measurements with modified skyradiometer
POM-02 using the moon as a light source
Authors:
Akihiro Uchiyama[1], Masataka Shiobara[2], Hiroshi Kobayashi[3], Tsuneo Matsunaga[1],
Akihiro Yamazaki[4], Kazunori Inei[5], Kazuhiro Kawai[5], and Yoshiaki Watanabe[5]
[1]Center for Global Environmental Research, National Institute for Environmental
Studies, Tsukuba, Ibaraki 305-8506, Japan
[2]National Institute of Polar Research, Midoricho, Tachikawa, Tokyo 190-8518, Japan
[3]University of Yamanashi, Takeda, Kofu, Yamanashi 400-8510, Japan
[4]Meteorological Research Institute, Japan Meteorological Agency, Tsukuba, Ibaraki
305-0052, Japan
[5]Prede Co., Ltd., Kamidaira, Fussa, Tokyo 197-0012, Japan
Correspondence to: Uchiyama Akihiro (uchiyama.akihiro@nies.go.jp)
**Abstract**
The majority of aerosol data are obtained from daytime measurements, and there are
few datasets available for studying nighttime aerosol characteristics. In order to
estimate the aerosol optical depth (AOD) and the precipitable water vapor (PWV)
during the nighttime using the moon as a light source, a skyradiometer POM-02
(Prede Ltd., Japan) was modified. The amplifier was adjusted so that POM-02 could
measure lower levels of input irradiance. In order to track the moon based on the
calculated values, a simplified formula was incorporated into the firmware. A new
position sensor with a four-quadrant detector to adjust tracking of the sun and the
moon was also developed.
The calibration constant, which is the sensor output for the extra-terrestrial solar
and lunar irradiance at the mean earth-sun distance, was determined by using the
Langley method. The measurements for the Langley calibration were conducted at
the NOAA/MLO in October and November 2017. By assuming that the relative
variation of the reflectance of the Robotic Lunar Observatory (ROLO) irradiance
model is correct, the calibration constant for the lunar direct irradiance was
successfully determined using the Langley method. The ratio of the calibration
constant for the moon to that for the sun was often greater than 1; the value of the





ratio was 0.95 to 1.18 in the visible near-infrared wavelength region. This means
that the ROLO model often underestimates the reflectance. In addition, this ratio
depended on the phase angle. In this study, this ratio was approximated by a
quadratic expression of the phase angle. By using this approximation, the reflectance
of the moon can be calculated to within an accuracy of 1% or less.
In order to validate the estimates of the AOD and PWV, continuous measurements
with POM-02 were conducted at MRI/JMA from January 2018 to May 2018, and the
AOD and PWV were estimated. The results were compared with the AOD and PWV
obtained by independent methods. The AOD was compared with that estimated from
NIES High Spectral Resolution Lidar measurements (wavelength: 532 nm), and the
PWV was compared with the PWV obtained from a radiosonde and the Global
Positioning System. As a result, the estimations of the AOD and the PWV using the
moon as the light source were made with the same degree of precision and accuracy
as the estimates using the sun as the light source.


1. Introduction

Atmospheric aerosols are an important constituent of the atmosphere. Aerosols
change the radiation budget directly by absorbing and scattering solar radiation and
indirectly through their role as cloud condensation nuclei (CCNs), thereby increasing
cloud reflectivity and lifetime (e.g., Ramanathan et al. 2001; Lohmann and Feichter
2005). Aerosols also affect human health as one of the main components of air
pollution (Dockery et al. 1993; WHO 2006, 2013).
Atmospheric aerosols have a large variability in time and space. Therefore,
measurement networks covering an extensive area on the ground and from space
have been developed and established to determine the spatiotemporal distribution of
aerosols. Well-known ground-based networks include AERONT (AErosol RObotic
NETwork) (Holben et al. 1998), SKYNET (Takamura et al. 2004), and PFR-GAW
(Precision Filter Radiometer-Global Atmosphere Watch) (Wehrli 2005). These
observation networks use passive radiometers which measure sunlight in the region
from ultraviolet to shortwave infrared wavelengths and the column average effective
aerosol characteristics such as aerosol optical depth (AOD) are retrieved.
Using lidar, which is an active remote sensing instrument, several networks have
also been constructed: for example, the Micropulse Lidar Network (MPLNET) by
NASA (National Aeronautics and Space Administration) (Welton et al. 2001; Levis et
al. 2016), the European Aerosol Research Lidar Network (EARLINET) (Pappalardo



et al. 2016) in Europe, the Asian Dust and aerosol lidar observation network
(AD-Net) (Shimizu et al. 2017) in east Asia, and the Latin American Lidar Network
(LALINET) (Guerrero-Rascado et al. 2016) in South America.
Several satellite programs provide aerosol optical depth data on a global scale: for
example, the Moderate Resolution Imaging Spectroradiometer (Remer et al. 2005),
Multiangle Imaging Spectroradiometer (Kahn et al. 2005), Geostationary
Operational Environmental Satellite (GOES) Aerosol Smoke Product (Prados et al.
2007), Sea-viewing Wide Field-of-view Sensor (Wang et al. 2000), Advanced
Himawari Imager (AHI) (Yoshida et al. 2018; Kikuchi et al. 2018), and Cloud-Aerosol
Lidar and Infrared Pathfinder Satellite Observation (CALIPSO; Winker et al. 2007).
With the exception of active sensor measurements such as lidar systems, to
estimate aerosol characteristics, direct solar irradiance and scattered solar radiance
measured with a passive sensor are required. Therefore, the majority of aerosol
property data are obtained by daytime measurements, and there are few datasets of
nighttime aerosol characteristics available.
To advance the understanding of the diurnal behavior of aerosols, and nocturnal
mixing layer dynamics, nighttime continuous AOD measurements are necessary. In
particular, in high latitude regions during the winter polar night, aerosol properties
cannot be measured using sunlight, and this results in gaps in the long-term aerosol
data. Such nocturnal aerosol data would also contribute to the understanding of
aerosol transport to polar regions, the influence of aerosol on cloud formation, and
the cloud effect on the radiation budget.
Lidar instruments can be used to obtain aerosol data during the night. However, in
many cases, a value for the ratio of the extinction coefficient to the backscattering
coefficient is often assumed in analysis using lidar, and in order to improve the
accuracy of the analysis, constraining of the AOD is necessary.
In order to measure the optical depth of aerosol at night, research has been
conducted using the moon and stars as light sources (Herber et al. 2002; Esposito et
al. 1998; Esposito et al. 2003; Pérez-Ramírez et al. 2008). Since the reflectivity of the
moon changes depending on the observation angle, the determination of the
calibration coefficient is an important obstacle to overcome (Herber et al. 2002).
Instruments for observing stars are large, expensive, and complicated to use due to
the low level of incoming energy from stars. Therefore, stellar measurements are
limited in use, and no large-scale observation network has been established.
The moon is a bright light source at night and the reflectance properties of the
moon's surface are virtually invariant ($<10^{-8}\,\text{yr}^{-1}$; Kieffer 1997). However, since the
surface of the moon is not spatially uniform and has non-Lambertian reflectance, the





brightness of the moon as seen by an observer on the earth varies depending on the relationship between the moon, the sun, and the observer, that is, the phase and the lunar libration. Therefore, it is difficult to use the moon as a light source.

However, in the 2000s, the quality of reflectance data for the moon has improved. The empirical model known as ROLO (Robotic Lunar Observatory) was developed by the United States Geological Survey (USGS) (Kieffer and Stone 2005). ROLO is a NASA-funded program aimed at using the moon for on-orbit calibration of Earth Observing System (EOS) satellite instruments. Furthermore, the Spectral Profiler (SP) onboard the Japanese Selenological and Engineering Explorer (SELENE, nicknamed Kaguya) measures lunar photometric properties in the region of visible, near-infrared, and shortwave infrared wavelengths (Yokota et al. 2011). These data made it possible to estimate the reflectance of the moon, and thus the moon can be used as a light source for aerosol optical depth estimation.

The Cimel sun photometer used in AERONET has been modified for lunar observation and the aerosol optical depth at night can be estimated (Berkoff et al. 2011; Barreto et al. 2013, 2016, 2017).

In SKYNET, the radiometers POM-01 and POM-02, manufactured by Prede Co. Ltd., Japan, are used. These radiometers are called 'sky radiometers', and measure both the solar direct irradiance and sky-radiances (Takamura et al. 2004). The sky radiometers POM-01 and POM-02 (Prede Co. Ltd., Japan) can measure solar direct irradiance and sky-radiances during the daytime and the measured data are used for estimating aerosol characteristics during the daytime (Takamura et al. 2004). In this study, we will aim to measure the optical depth of aerosol using the moon as a light source by modifying POM-02.

In section 2, we describe our modification of the instrument. In section 3, the ROLO model is briefly explained. In section 4, we briefly describe the data used in this study. In section 5, the calibration method and corresponding results are described. In section 6, we show the results of comparing the aerosol optical thickness and precipitable water vapor obtained by continuous observation with those obtained by other independent instruments.

2. Modification of instrument

2.1 Adjustment of Amplifier

The skyradiometer POM-02 is designed to measure the direct solar irradiance and the scattered sky radiance with a single radiometer. An example of the calibration constant, which is the sensor output for the extra-terrestrial solar irradiance at the



mean earth-sun distance (1 astronomical unit (AU)) at the reference temperature, is
shown in Table 1. It is $1.8 \times 10^{-5}$ to $3.4 \times 10^{-4}$ A in the visible and near-infrared regions,
and $7.9 \times 10^{-5}$ to $1.3 \times 10^{-4}$ A in the short-wavelength infrared region. Figure 1 shows
an example of measurements of scattered radiances in the visible and near-infrared
wavelength region. The output for the scattered radiance from the sky is $1 \times 10^{-7}$ to
$1 \times 10^{-10}$ A, and this value is $1 \times 10^{-6}$ smaller than the direct solar irradiance. The
direct lunar irradiance is $1 \times 10^{-5}$ as strong as the direct solar irradiance during a full
moon, and $1 \times 10^{-6}$ during a half-moon (Berkoff et al. 2011). The direct lunar
irradiance during the half moon is about $2 \times 10^{-5} \times 10^{-6} = 2 \times 10^{-11}$ in the 340 and 380
nm channels. This is nearly detectable limits of the current POM-02. Without the
modification, it is possible to measure the direct lunar irradiance with the current
POM-02 except for wavelengths between 340 and 380 nm where the sensitivity of the
detector is low and wavelengths of 1225, 1627, and 2200 nm with poor S/N.
Table 2 shows the measurement range before and after modification of POM-02.
POM-02 measures input energy in seven ranges according to the magnitude of the
input energy, and the measured value is digitized with 15 bits. The measurement
range was expanded slightly. The measurement limit depends on the magnitude of
the dark current and the magnitude of the noise.
The dark current of the detector in the visible and near-infrared region was about
$5 \times 10^{-13}$ A, and the RMS of the random component of the noise was $4 \times 10^{-14}$ A. In
consideration of these values, the new POM-02 can use the amplifiers from range 1
to 7 and the minimum meaningful current is about $4 \times 10^{-13}$ A (~RMS$\times$10) in the
visible and near-infrared region. This is $1 \times 10^{-8}$ to $1 \times 10^{-9}$ of the direct solar
irradiance.
The dark current of the detector in the shortwave infrared wavelength region was
about $1.5 \times 10^{-8}$ A, and the RMS of the random component of the noise was $4 \times 10^{-11}$ A.
The new POM-02 can use the amplifiers from range 1 to 5 and the minimum
meaningful current is about $4 \times 10^{-10}$ A (~RMS$\times$10). This value and the magnitude of
the measured value of the direct lunar irradiance are comparable. It is difficult to
measure direct lunar irradiance even with the new POM-02 in the shortwave
infrared wavelength region.

2.2 Sun and moon position sensor

A position sensor with a four-quadrant detector is used to adjust the tracking of the
sun and the moon. In order to adjust the tracking of the moon, a position sensor
incorporating a new electronic circuit to amplify the signal and new software to





process the signal data was developed. The new position sensor can be used to track
both the sun and the moon. The function of the moon tracking adjustment works for
a period of the full moon ± about 90 degrees of the phase angle (half-moon). The
threshold value to operate the tracking adjustment can be specified by the user. For
phase angles larger than the half-moon, the signal of the position sensor was small,
and the position sensor was not used. When the POM-02 is accurately installed
horizontally and directionally, it is possible to track the moon based on the calculated
position values, and the direct irradiance of the moon could be measured during the
full moon to within ± about 10 days, depending on the aerosol optical depth on the
measurement day.

2.3 Simplified calculation of moon position

The moon positions were calculated with simplified formula in Nagasawa (1981).
When comparing the moon position calculated by this simplified formula with that
calculated using the NASA SPICE toolkit (Acton 1996), the difference in the zenith
angle is less than 0.01 degrees, and the difference in the azimuth angle is less than
0.04 degrees. Therefore, if POM-02 is accurately installed horizontally and
directionally, it is possible to track the moon with sufficient accuracy based on values
calculated with the simplified formula.

3. Robotic Lunar Observatory (ROLO) irradiance model

In order to estimate the aerosol optical thickness using the moon as a light source,
measurement of the extra-terrestrial irradiance of the moon is necessary. In this
study, a model known as the ROLO irradiance model (Kieffer and Stone 2005) was
used. This model was developed at the U.S. Geological Survey (USGS) and is based
on an extensive database of radiance images acquired by the ground-based ROLO
over more than 8 years. ROLO is a NASA-funded program designed to use the moon
for on-orbit calibration of Earth Observing System (EOS) satellite instruments. The
empirical irradiance model was developed for 32 wavelengths from 350 to 2450 nm
and has the same form for each wavelength. The average residual is less than 1%.
The coefficients of the empirical formula were constrained and determined using
data with a phase angle between 1.55 and 97 degrees.



$$\ln A_k = \sum_{i=0}^{3} a_{ik} g^i + \sum_{j=1}^{3} b_{jk} \Phi^{2j-1} + c_1 \phi + c_2 \theta + c_3 \Phi \phi + c_4 \Phi \theta$$
$$+ d_{1k} e^{-g/p_1} + d_{2k} e^{-g/p_2} + d_{3k} \cos((g - p_3)/p_4)$$
(1)

where $A_k$ is the disk-equivalent reflectance, $g$ is the absolute phase angle in
radians, $\theta$ and $\phi$ are the selenographic latitude and longitude of the observer in
degrees, and $\Phi$ is the selenographic longitude of the sun in radians.
This formula must be used with caution. The equation in Kieffer and Stone (2005)
has well-known typographical errors. In eq. (1), $\theta$ and $\phi$ in the original expression
by Kieffer and Stone (2005) are exchanged. In addition, the units of the coefficients
$p_1$, $p_2$, $p_3$, and $p_4$ are degrees. Therefore, in order to make the dimensions the
same, $g$ in the exponent and the cosine terms must be converted into units in
degrees.
The astronomical parameter was calculated using our own software developed
using the NASA SPICE toolkit; an observation geometry information system named
SPICE is offered by NASA's Navigation and Ancillary Information Facility (NAIF)
(Acton, 1996). SPICE is widely used in the NASA and international planetary
exploration communities (for more information about SPICE, refer to the NAIF web
page at http://naif.jpl.nasa.gov.).
In this study, only the values of the reflectance are used, and it is assumed that the
relative changes in reflectance are correct. The reflectance values are not converted
to irradiance values by assuming the extra-terrestrial solar spectral irradiance.
The wavelength of POM-02 used in this study does not necessarily match the
wavelength of the ROLO model. Here, the reflectance at the wavelength of POM-02
was calculated by linearly interpolating from the reflectance of the ROLO model at
two adjacent wavelengths. In addition, the ROLO model does not have reflectance
data for the wavelength 340 nm. The reflectance at the wavelength 340 nm was
obtained by extrapolating linearly from the values at the two end wavelengths.

4. Data
4.1 Data for Langley calibration

The aerosol optical thickness is estimated by measuring the attenuation of the
direct solar or lunar irradiance. Therefore, in order to estimate the aerosol optical
thickness, the output of the instrument for the input irradiance at the top of the





atmosphere is necessary. The determination of this constant is referred to as
calibration, and the output of the instrument for the extra-terrestrial solar or lunar
irradiance at the mean earth-sun distance (1 AU) at the reference temperature is
called the calibration constant. In this study, the calibration constant was
determined by the Langley method.
To calibrate the POM-02 by the Langley method, measurements were conducted
at the NOAA Mauna Loa Observatory (MLO) during the period from Sep. 28, 2017 to
Nov. 7, 2017; the full moon was on Oct. 4 and Nov. 3, 2017. The MLO (19.5362°N,
155.5763°W) is located at an elevation of 3397.0 meters amsl on the northern slope of
Mauna Loa, Island of Hawaii, Hawaii, USA. The atmospheric pressure is about 680
hPa. The MLO is one of the most suitable places to obtain data for a Langley plot for
the solar direct irradiance measurement (Shaw 1983). Though the air at MLO is
highly transparent, it is affected in the late morning and afternoon hours by marine
aerosol that reaches the observatory during the marine inversion boundary layer
breakdown under solar heating (Shaw 1983; Perry et al. 1999). Therefore, using data
taken in the morning is recommended (Shaw 1982; Dutton et al. 1994; Holben et al.
268 1998).

However, during the nighttime, the upslope winds change to downslope winds,
which bring low moisture and aerosol-poor air above the marine boundary layer
down to the observatory. As a result, daytime orographic clouds at the observatory
disappear and the atmosphere stratification becomes stable. These atmospheric
conditions are suitable for obtaining data for the Langley plot from the lunar direct
irradiance measurement.

4.2 Continuous measurement for comparison

The measurements for the estimation of the aerosol optical depth and precipitable
water vapor were performed at 1-minute intervals at the Japan Meteorological
Agency/Meteorological Research Institute (JMA/MRI) (36.05°N, 140.13°E) in
Tsukuba, which is located about 50 km northeast of Tokyo. The comparison was
made using data obtained during the period from Jan. 1 to May 31, 2018. During this
period, the AOD and the precipitable water vapor (PWV) were estimated assuming
the calibration constant was unchanged.
The optical depth estimated from POM-02 was compared with the value of the
NIES High Spectral Resolution Lidar (HSRL, wavelength; 532 nm). The NIES/HSRL
is one of the lidar operated by the lidar measurement group of the National Institute
for Environmental Studies (NIES) (Shimizu et al. 2016). The NIES and MRI





observation sites are located about 800 m apart. Since the POM-02 was not
measured at the 532 nm wavelength, the AOD at 532 nm was interpolated from the
values of 500 nm and 675 nm by assuming that AOD is proportional to $\lambda^{-\alpha}$, where
$\lambda$ is the wavelength. Furthermore, since the AOD of NIES/HSRL is the 15-minute
average, the value of POM-02 was also averaged over 15 minutes.
The PWV estimated from POM-02 was compared with that obtained from the
vertical profile of a radiosonde and that obtained from the Global Positioning System
(GPS) receiver. The radiosonde observation is operated from the JMA Aerological
Observatory, which is adjacent to JMA/MRI. The GPS receiver is installed at
JMA/MRI, and GPS data were processed by one of the JMA/MRI researchers (Shoji
et al. 2013). The comparison of the PWV was performed using the 30-minute average
values.

5. Calibration of POM-02 using MLO data
5.1 Method

The AOD and PWV are estimated by measuring the attenuation of the direct solar or
lunar irradiances. Therefore, it is necessary to know the output of the instrument for
extra-terrestrial solar or lunar irradiances. This output at the mean earth-sun
distance (1 AU) at the reference temperature is called the calibration constant. In
this study, the calibration constant was determined by the Langley method
(Uchiyama et al. 2018). Here, we do not consider the temperature dependence of the
sensor output for the POM-02. Under these observation conditions in Tsukuba, the
temperature dependence of the sensor output can be ignored except for the 340, 380,
and 2200 nm channels (Uchiyama et al. 2018).
The sensor output when measuring the direct solar irradiance can be written as
follows:
$$V(\lambda_0) = \frac{V_{S0}(\lambda_0)}{R_S^2} \exp(-m(\theta)\tau(\lambda_0))\overline{T}_{gas}(\lambda_0,\theta) \qquad (2)$$
where $V(\lambda_0)$ is the sensor output in the $\lambda_0$ wavelength channel, $R_S$ is the
earth-sun distance in AU, $m(\theta)$ is the total airmass, $\tau(\lambda)$ is the total optical
depth, $\theta$ is the solar zenith angle, and $\overline{T}_{gas}(\lambda_0,\theta)$ is the channel average
transmittance of the gas line absorption. Furthermore, $V_{S0}(\lambda_0)$ is the sensor output
for the extra-terrestrial solar irradiance at 1 AU, and is called the calibration





constant. $\tau(\lambda)$ consists of the optical thickness for molecular scattering (Rayleigh
scattering), aerosol, and the continuous absorption of gas. In this study, it is assumed
that airmass $m(\theta)$ is the same for all components. The airmass $m(\theta)$ for
molecular scattering is often used (Schmid and Wehrli 1995; Holben et al. 1998).

In the case of no "gas absorption", the following equation is used:

$$V(\lambda_0) = \frac{V_{S0}(\lambda_0)}{R_S^2} \exp(-m(\theta)\tau(\lambda_0)) \qquad (3)$$
Taking the logarithm of the equation leads to
$$\ln(V(\lambda_0)R_S^2) = \ln V_{S0}(\lambda_0) - m(\theta)\tau(\lambda_0)$$
$$= C_1 m(\theta) + C_2 \qquad (4)$$

The parameters on the left-hand side are known: $V$ is the measurement value, and
$R_S$ and $m(\theta)$ can be calculated from the solar zenith angle. For example, $R_S$ can
be calculated with the simplified formula in Nagasawa (1981), and $m(\theta)$ can be
calculated as in Kasten and Young (1989). In the case of POM-02, the sensor output
is the current, and the unit of the measurements of $V$ is the ampere A. $C_2 = \ln V_{S0}$
is determined from the ordinate intercept of a least-square fit when one plots the
left-hand side of the above equation versus airmass $m(\theta)$.

For the water vapor absorption band at a wavelength of 940 nm, the

Beer-Lambert-Bouguer law is not valid. Calibration methods for the 940 nm channel,
which is in the water vapor absorption band, have been considered extensively in
previous studies (Reagan et al. 1987a, 1987b, 1995; Bruegge et al. 1992; Thome et al.
1992, 1994; Michalsky et al. 1995, 2001; Schmid et al. 1996, 2001; Shiobara et al.
1996; Halthore et al. 1997; Cachorro et al. 1998; Plana-Fattori et al. 1998, 2004;
Ingold et al. 2000; Kiedron et al. 2001, 2003; Uchiyama et al. 2014, Campanelli et al.

2014).

In this study, the modified Langley method is used (Reagan et al. 1987a; Bruegge

et al. 1992; Schmid and Wehrli 1995). In the modified Langley method, the
transmittance is approximated by an empirical formula. The water vapor
transmittance is approximated as follows:
$$Tr(\text{H2O}) = \exp(-a(m(\theta) \cdot pwv)^b) \qquad (5)$$
where $a$ and $b$ are fitting coefficients, and $pwv$ is PWV.
Coefficients $a$ and $b$ were determined by computing the transmittance for several
atmospheric models (Uchiyama et al. 2014).





The output of the 940 nm channel can be written as follows;

$$
V(\lambda_0) = \frac{V_{S0}(\lambda_0)}{R_S^{\ 2}} \exp(-m(\theta)\tau(\lambda_0)) Tr(\mathrm{H2O})
$$
$$
= \frac{V_{S0}(\lambda_0)}{R_S^{\ 2}} \exp(-m(\theta)\tau(\lambda_0)) \exp(-a(m(\theta)\cdot pwv)^b)
$$

(6)

Taking the logarithm of the equation leads to

$$
\ln V R_s^{\ 2} + m(\theta)(\tau_{aer} + \tau_R) = \ln V_{S0} - a(pwv)^b m(\theta)^b
$$
$$
= C_1 m(\theta)^b + C_2
$$

(7)

In the same way as the normal Langley method, the parameters on the left-hand side
are known: $V$ is the measurement value, and $R$ and $m(\theta)$ can be calculated from
the solar zenith angle. $\tau_R$ is also estimated from the surface pressure; for example,
$\tau_R$ can be calculated as in Asano et al. (1983). In addition, $\tau_{aer}$ is the aerosol optical
depth at the 940 nm wavelength, which is interpolated from the aerosol optical depth
from the values at the 870 and 1020 nm wavelengths.
If $pwv$ is constant, then the right-hand side of the equation is a linear function of
$m(\theta)^b$. Therefore, the values on the left-hand side can be fitted by a linear function
of $m(\theta)^b$, and the intersection of the y-axis and the fitted line is $\ln V_{S0}$.

The sensor output when measuring the direct lunar irradiance can be written as
follows:
$$ V(\lambda_0) = \frac{A_{ROLO}}{\pi} \Omega_M \frac{V_{S0}(\lambda_0)}{R_S^{\ 2}} \cdot \frac{1}{R_m^{\ 2}} \exp(-m(\theta)\tau(\lambda_0)) \overline{T}_{gas}(\lambda_0, \theta) $$ (8)
where $A_{ROLO}$ is the lunar reflectance by the ROLO irradiance model, $\Omega_M$ is the
solid angle of the moon, $R_S$ is the distance between the moon and the sun in AU,
and $R_m$ is the distance between the moon and the observer in AU.

It is assumed that the relative variation of the reflection of the ROLO model is
correct. This assumption means that the reflectance of the moon is assumed to be
$C \cdot A_{ROLO}$, where $C$ is a constant. $A_{ROLO}$ in eq. (8) is replaced with $C \cdot A_{ROLO}$.



$$V(\lambda_0) = \frac{C \cdot A_{ROLO}}{\pi} \Omega_M \frac{V_{S0}(\lambda_0)}{R_S^{\,2}} \cdot \frac{1}{R_m^{\,2}} \exp(-m(\theta)\tau(\lambda_0))\bar{T}_{gas}(\lambda_0,\theta) \qquad (9)$$
In the case of no "gas absorption", taking the logarithm of the equation leads to
$$\ln(\frac{\pi V(\lambda_0)}{A_{ROLO}\Omega_M} R_S^{\,2} R_m^{\,2}) = \ln C V_{S0}(\lambda_0) - m(\theta)\tau(\lambda_0)$$
$$= \ln V_{m0}(\lambda_0) - m(\theta)\tau(\lambda_0) \qquad (10)$$
$$= C_1 m(\theta) + C_2$$

where $V_{m0}(\lambda_0) = C V_{S0}(\lambda_0)$. $C_2 = \ln V_{m0}$ is determined from the ordinate intercept
of a least-square fit when one plots the left-hand side of the above equation versus
airmass $m(\theta)$.
$\quad$ $V_{S0}$ can be determined by applying the Langley method to data taken during the
daytime. If $V_{S0}$ is determined, the coefficient $C$ can be determined by taking the
ratio of $V_{m0}$ and $V_{S0}$. If the coefficient $C$ is 1, the reflectance of the ROLO model
will be correct. If the coefficient $C$ is greater than 1 (less than 1), the reflectance in
the ROLO model is under-estimated (over-estimated).
5.2 Results
$\quad$ Examples of Langley plots in the visible and near-infrared wavelengths are shown
in Fig. 2. In these examples, the regression lines can be well determined for any
wavelength. $C_2 = \ln V_{m0}$ is determined from the ordinate intercept of the regression
line (see eq. (10)). At the 340 nm wavelength, the regression line tends to deviate
from the measured values in the region of airmasses larger than 6. It is presumed
that the detector output at the 340 nm wavelength is small and hence the detector
output may be nonlinear. The output at that time is about $1\times10^{-12}$ A. When using
output values less than this, the user needs to treat there results with caution. At the
940 nm wavelength, the modified Langley method was applied. In this example, the
regression line equation provides a good fit.
$\quad$ In Fig. 3, examples of the Langley plot in the shortwave infrared region (1225, 1627,
2200 nm) are shown. The detector output of these channels are from $2\times10^{-10}$ to





5×10⁻¹⁰ A, and the root mean square error of the random noise is 4×10⁻¹¹ A. The ratio
of noise to detector output is large and it is difficult to use these channels for
estimating the aerosol optical depth.
In Fig. 4, the relationship between the coefficient $C(=V_{m0}/V_{S0})$ and the phase
angle in the visible and near-infrared wavelength region (from 340 to 1020 nm) is
shown. As shown in the previous section, the coefficient $C$ is the ratio of the
calibration constant for the moon and sun, which indicates the error of the ROLO
irradiance model reflectance and thus the ROLO irradiance model reflectance can be
corrected using the coefficient $C$. As can be seen from this figure, the coefficient $C$
is often greater than 1 and depends on the phase angle. At most wavelengths, the
coefficient $C$ is small when the absolute value of the phase angle is small (near the
full moon) and increases as the absolute value of the phase angle increases. The
range of $C$ is 0.95 to 1.18. The absorption band of water vapor is at the 940 nm
wavelength, and the accuracy of both $V_{S0}$ and $V_{m0}$ is poor. Therefore, no clear
relationship between $C$ and the phase angle is found, but the coefficient $C$ is
about 1.16. The fact that $C$ is larger than 1 means that the reflectance of the ROLO
irradiance model is underestimated.
In Fig. 5, the relationship between the coefficient $C(=V_{m0}/V_{S0})$ and the phase
angle in the shortwave infrared wavelength region (1225, 1627, 2200 nm) is shown.
In these channels, the error of $C$ is large, but the coefficient $C$ depends on the
phase angle as in the visible and near-infrared wavelength region; $C$ is small when
the phase angle is near zero and increases as the absolute value of the phase angle
increases.
In this study, the phase angle dependence of the coefficient $C$ is approximated by
a quadratic equation of the absolute value of the phase angle:
$C = A_c \cdot g^2 + B_c$ (11)
where $g$ is the phase angle.
That is,
$V_{m0} = V_{S0} \cdot (A_c \cdot g^2 + B_c)$ (12)
The coefficients, $A_c$ and $B_c$ are shown in Table 3. The regression line was plotted
in Figs. 4 and 5. By using this approximation, the reflectance of the ROLO model can
be estimated to within 1% in most channels. By using this approximation, the data
processing to estimate the aerosol optical depth from the measured value becomes
straightforward.
The error of the reflectance in the ROLO irradiance model is dependent on the
phase angle. Here, although it was approximated by the second-order regression
equation, there still remains error beyond the uncertainty of the Langley plot. These
errors may be caused by the effect of the libration. It is necessary to further
accumulate the reflectance data of the moon.

6. Examples of measurement

In order to validate the estimations of AOD and PWV, we compared them with the
AOD and PWV obtained by independent methods.

6.1 Aerosol optical depth (AOD)

The AOD estimated from POM-02 was compared with the value of the NIES/HSRL
(wavelength: 532 nm). The AOD at 532 nm was interpolated from the values for 500
and 675 nm by assuming that the AOD is proportional to $\lambda^{-\alpha}$, where $\lambda$ is
wavelength.

Figures 6 (a) and (b) show the scatter plot of the aerosol optical depth during the

daytime and nighttime, respectively. In Fig. 6 (c), the scatter plot during the
nighttime is shown together with that during the daytime. Table 4 shows the results
of the comparison between NIES/HSRL and POM-02 AOD: the statistics of the
difference between the two AODs, the coefficients of the linear regression equation of
NIES/HSRL and POM-02 AOD ($\tau_{HSRL} = C_1 \cdot \tau_{POM02} + C_2$ .), the RMSE of the residual,
the 95% confidence interval of the coefficients, and the number of observations.

The difference in the slope value of the regression coefficients is 0.1600 (= 1.0477 −

0.8877). The 95% confidence interval of the coefficient is about ±0.04 during both the
daytime and the nighttime. It cannot be said that the slopes of the two regression
lines are equal based on their 95% confidence intervals. However, the correlation
between NIES/HSRL and POM-02 AOD is high, and the difference between them
and their RMSEs are similar. Furthermore, as shown in Fig. 6 (c), the scatter
diagrams for the daytime and nighttime are almost overlapping, and it seems that
the two sets of measurements obtain similar results.

Examples of time series of the AOD from NIES/HSRL and POM-02 are shown in





Fig. 7. As can be seen from these figures, the AOD of the daytime and nighttime
estimated from POM-02 constitute a continuous series. The AOD from NIES/HSRL
and that from POM-02 have similar time variations. However, while there are
periods when the values are consistent, there are periods when there are systematic
differences.
In NIES/HSRL data processing, the AOD below an altitude of 500 m is calculated
by using the value of the extinction coefficient for an altitude of 500 m. Since the
height of the atmospheric boundary layer is typically 1500 to 2000 m, a large amount
of aerosols exist at altitudes below 500 m. If the actual distribution deviates from the
assumed distribution, the estimated AOD is shifted systematically.
6.2 Precipitable water vapor (PWV)
The PWV estimated from POM-02 was compared with that obtained from the
vertical profile of the radiosonde and that obtained from the GPS receiver. The PWV
estimated from the radiosonde data has a frequency of two values per day, whereas
the PWV obtained from GPS is continuous.
6.2.1 Radiosonde
The PWV from a radiosonde is often used as a reference for the PWV measurement
value. The PWV from the radiosonde and PWV from POM-02 are first compared.
Figure 8 shows a scatter plot of the PWV from the radiosonde and from POM-02. The
red symbol denotes 00 UTC (09 LTC), and the blue symbol is 12 UCT (21 LTC). Table
5 shows the results of the comparison between the radiosonde and POM-02
precipitable water vapor: statistics of the difference between both PWV, the
coefficients of the linear regression equation of the radiosonde and POM-02 PWV
($PWV_{POM-02} = C_1 \cdot PWV_{Sonde} + C_2$), the RMSE of residual, 95% confidence interval of
the coefficients, and number of data. The ratio of PWV estimated from POM-02 and
the radiosonde in both daytime and nighttime is almost constant: the slope of the
regression line is 0.80 in the daytime and 0.78 in the nighttime.
The empirical formula of the transmittance is expressed as eq. (5). The ratio of the
two PWVs is almost constant. In addition, as shown in Fig. 4, the modified Langley
plot provides a good fit for the data. From these facts, it seems that the value of the
coefficient $b$ of eq. (5) is appropriate but the value of the coefficient $a$ of eq. (5)
was inappropriate. It is possible that the filter characteristics of the 940 nm channel





have changed from the nominal characteristics due to degradation.
Let $pwv = c \cdot pwv'$ and rewrite eq. (5) as follows:
$$Tr(\text{H2O}) = \exp(-a(m(\theta) \cdot (c \cdot pwv'))^b)$$
$$= \exp(-ac^b(m(\theta) \cdot pwv')^b)$$
(13)

Then the PWV can be corrected by replacing $a$ with $ac^b$.
Figure 9 shows a scatter plot of the PWV from the radiosonde and the corrected
PWV from POM-02. For the correction coefficient $c$, the average value of the
coefficients $C_1$ of the daytime and nighttime regression equations was used. Table 6
shows the results of the comparison between the radiosonde and corrected POM-02
precipitable water vapor: the statistics of the difference between both PWV, the
coefficients        of        the        linear        regression        equation
$PWV_{POM-02}(\text{corrected}) = C_1 \cdot PWV_{Sonde} + C_2$, the RMSE of the residual, the 95%
confidence interval of coefficients, and the number of observations.
The slope $C_1$ of the regression line during the daytime and nighttime is 1.0160
and 0.9869, respectively, and the difference between them is 0.0291 (= 1.0160 −
0.9869). The 95% confidence intervals of the slopes during the daytime and
nighttime are ± 0.0206 and ± 0.0271, respectively. The difference between them is
0.0291, which is larger than the respective 95% confidence intervals. Therefore, the
two slopes are not equivalent based on the 95% confidence intervals.
However, since the slope of the regression line determined using all of the data is
1.0042 and the 95% confidence interval is ±0.0173, the three slopes of the regression
lines can be regarded as equivalent at the 95% confidence level. Furthermore, there
are no large differences in the bias, RMSE, and correlation coefficient between PWV
from the radiosonde and POM-02. Therefore, the PWVs of daytime and nighttime for
POM-02 are presumed to have the same degree of precision and accuracy.

6.2.2 GPS

Next, the result of the comparison between the PWV obtained from POM-02 and
GPS is shown. Before that, the result of the comparison between the PWV obtained
from GPS and the radiosonde is shown in Fig. 10. Table 7 shows the results of the
comparison between GPS and radiosonde precipitable water vapor: the statistics of
the difference between both PWV, the coefficients of the linear regression equation


$PWV_{Sonde} = C_1 \cdot PWV_{GPS} + C_2$, the RMSE of the residual, the 95% confidence interval
of the coefficients, and the number of observations.
The slope of the regression line in Fig. 10 is about 0.94. In the region of the PWV
less than 2 g/cm², the PWV from GPS tends to be smaller than the PWV from the
radiosonde. In the region of PWV more than 3 g/cm², the difference between PWV
from GPS and the radiosonde is more scattered. Therefore, the slope of the
regression line became smaller than 1. In a previous comparison conducted by the
authors, the slope of the regression line was almost 1 (Uchiyama et al. 2014). There
is a possibility that the PWV from GPS used in this study has a larger error than the
PWV used previously.
Figure 11 shows a scatter diagram of the PWV from GPS and the corrected PWV
from POM-02. Table 8 shows the results of the comparison between PWV from GPS
and corrected PWV from POM-02: statistics of difference between the two PWVs, the
coefficients of the linear regression equation, the RMSE of the residuals, the 95%
confidence interval of the coefficients, and the number of observations.
The slope of the regression line is about 0.91 for both the daytime and nighttime.
Similar to the results of the comparison between the PWV from the radiosonde and
GPS, in the region of PWV from GPS less than 2 g/cm², the PWV from GPS tends to
be somewhat smaller than the PWV from POM-02 during both the daytime and
nighttime. In the region of PWV greater than 3 g/cm², the difference between PWV
from GPS and the radiosonde is more scattered.
The difference between the slopes of the regression lines is 0.0076 (= 0.9132 −
0.9056) and the 95% confidence intervals during the daytime and nighttime are
±0.0097 and ±0.0221, respectively. Therefore, the confidence intervals of the two
slopes are overlapping, and the values of slopes can be regarded as equivalent at the
95% confidence level.
In Fig. 11 (c), the scatter plot obtained using nighttime data is shown together
with that obtained using daytime data. The data obtained during the daytime and
nighttime overlap, and it seems that the PWV from POM-02 during the daytime and
nighttime are estimated with the same degree of precision and accuracy.
Examples of time series of PWV from GPS and POM-02 are shown in Fig. 12. The
PWV from GPS and that from POM-02 have similar time variations. Although there
are some systematic differences in Fig. 12 (b), the PWV from GPS and the PWV from
POM-02 almost overlap in Figs. 12 (a) and (c). In addition, the PWV during the
daytime and nighttime estimated from POM-02 are continuously connected.



7. Summary and conclusion

Aerosol data are often estimated using the solar direct irradiance and the solar
scattered radiance. Therefore, the majority of data on aerosol properties are obtained
using daytime measurements, and there are few data available on aerosol
characteristics at night. In order to estimate the aerosol optical depth (AOD) and the
precipitable water vapor (PWV) during the nighttime using the moon as a light
source, POM-02 (Prede Ltd., Japan), which is used to estimate aerosol characteristics
during the daytime, was modified.
The current POM-02 has the ability to measure the direct irradiance from the
moon for some channels in the visible and near-infrared wavelength region without
the modification. Several modifications were made to measure the AOD during the
nighttime and the measurement range was expanded slightly.
The amplifier was adjusted so that POM-02 could measure up to about $5 \times 10^{-13}$ A:
the lunar direct irradiance can be measured in the wavelength range of 340 to 1020
nm.
In order to track the moon based on the calculated value, the simplified formula
by Nagasawa (1981) was incorporated into the firmware.
A position sensor with a four-quadrant detector is used to adjust the tracking of
the sun and the moon. In order to adjust the tracking of the moon, a position sensor
incorporating a new electronic circuit to amplify the signal and new software to
process the signal data were developed. The new position sensor can be used to track
both the sun and the moon.
The calibration constant was determined by using the Langley method. The
measurements of the solar and lunar direct irradiance were conducted at the
NOAA/MLO during the period from Sep. 28 to Nov. 7, 2017. Assuming that the
relative variation of the reflectance of the ROLO irradiance model is correct, the
calibration constant for the lunar direct irradiance was determined by using the
Langley method. The calibration by the Langley method was successfully performed.
The ratio of the calibration constant for the moon to that for the sun was often
greater than 1. This ratio shows the error of the ROLO irradiance model: the error of
the moon reflectance. The value of the ratio was 0.95 to 1.18 in the visible
near-infrared wavelength region. This means that the ROLO model often
underestimates the reflectance. In addition, this ratio depended on the phase angle:
when the phase angle was small (near the full moon), the ratio was small, and as the
phase angle became larger, the ratio increased. In this study, this ratio was
approximated by the quadratic expression of the phase angle. By using this
approximation, the reflectance of the moon can be calculated to within an accuracy of
1% or less.

The continuous measurement of POM-02 was conducted at MRI/JMA from
January 2018 to May 2018, and the AOD and PWV were estimated. In order to
validate the estimates of the AOD and PWV, we compared them with the AOD and
PWV obtained by independent methods. The AOD was compared with the AOD (532
nm) estimated from NIES/HSRL, and the PWV was compared with the PWV from
the radiosonde and GPS.

Concerning the AOD, there were sometimes systematic differences between
NIES/HSRL and POM-02. The cause of the systematic difference seems to be that
NIES/HSRL assumes a constant extinction coefficient at altitudes of less than 500 m.
The slopes of the linear regression lines during the daytime and nighttime could not
be said to be equivalent at the 95% confidence level, but the scatter diagrams of the
daytime and nighttime were almost overlapping. Therefore, the nighttime AOT was
estimated with the same degree of precision and accuracy as the daytime AOT.

Concerning the PWV, the slopes of the linear regression lines during the daytime
and nighttime were equivalent at the 95% confidence level in the comparisons
between the PWV from POM-02 and the radiosonde and in the comparison between
the PWV from POM-02 and GPS. Furthermore, the scatter diagrams of the daytime
and the nighttime data were almost overlapping. Therefore, the PWV at nighttime
can be estimated with the same degree of precision and accuracy as daytime PWV.

From these facts, the estimation of the AOD and the PWV using the moon as the
light source has the same degree of precision and accuracy as the estimation using
the sun as the light source.

In this study, the calibration was performed using about 40 days of data including
two full moon days. As a result, it was found that there was an error in the
reflectance of the ROLO irradiance model. In the future, it is necessary to
accumulate more data for calibration and to reduce the error of the ROLO irradiance
model. It is said that the ROLO model can be applied over a phase angle range of
about 90 degrees. POM-02 has the ability to measure the direct lunar irradiance up
to a phase angle range of about 120 degrees. It is necessary to expand the ROLO
irradiance model so that it can be applied to larger phase angles.

It is now possible to estimate the aerosol optical depth during the nighttime. It is
necessary to promote the adoption of this system in the existing observation network.
After that, it is necessary to use the data obtained by using this instrument in the
field of aerosol science at night, for the validation of aerosol transport models, and
input data of the assimilation system.





**Data availability.**
Data used in this study are available from the corresponding author.
**Authors contributions.**
This study was designed by AU, MS, HK and TM. The measurements of sky
radiometer were conducted by AU, AK, KI and YW. The adjustment of amplifier and
the development of position sensor were performed by MS, HK, KI, KK and YW. The
development of the related software and the data analyses were performed by AU.
The manuscript was written by AU, and all authors contributed to editing and
revision.
**Competing interests.**
The authors declare that they have no conflict of interest.
**Acknowledgements**
This work was supported by the NIES GOSAT-2 project, Japan. This work was also
supported by JSPS KAKENHI Grant Number 17K00531. We would like to thank Dr.
Y. Jin and Dr. T. Nishizawa of NIES for providing NIES/HSRL data for comparison of
the aerosol optical depth. We also would like to thank Dr. Y. Shoji of JMA/MRI for
providing GPS data for the comparison of precipitable water vapor.

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

Table 1 Examples of calibration coefficient $V_{S0}$ for the solar measurement
Table 2 Measurement range before (current) and after modification (new) of POM-02.
$I_n$ and $I_{n-1}$ are the upper and lower limit current (unit: A), respectively.

Table 3 Coefficients of the regression equation for reflectance correction factor C

Table 4 Results of comparison between NIES/HSRL and POM-02 aerosol optical
depth

Table 5 Same as Table 4 except for radiosonde and POM-02 precipitable water vapor

Table 6 Same as Table 4 except for radiosonde and corrected POM-02 precipitable
water vapor.

Table 7 Same as Table 4 except for GPS and radiosonde precipitable water vapor.

Table 8 Same as Table 4 except for GPS and corrected POM-02 precipitable water
vapor.

Figure captions
Fig. 1 An example of sensor output for the solar direct irradiances and the scattered
sky radiances by POM-02.

Fig. 2 Examples of Langley plot in the visible and near-infrared region.

Fig. 3 Examples of Langley plot in the shortwave infrared region.






Fig. 4 Relationship between phase angle and reflectance correction factor
$C = V_{m0} / V_{S0}$ in the visible and near-infrared region. A regression curve
($C = A_c \cdot g^2 + B_c$, $g$ : Phase angle) was also plotted.

Fig. 5 Relationship between phase angle and reflectance correction factor
$C = V_{m0} / V_{S0}$ in the shortwave infrared region. A regression curve ($C = A_c \cdot g^2 + B_c$, $g$ :
Phase angle) was also plotted.

Fig. 6 Scatter plot of HSRL and POM-02 aerosol optical depth. (a) daytime (red), (b)
nighttime (blue), (c) overlapping daytime (red) with nighttime (blue).

Fig. 7 Examples of time series of HSRL (red), POM-02 daytime (green) and
nighttime (blue) aerosol optical depths.

Fig. 8 Scatter plot of Radiosonde and POM-02 precipitable water vapor. daytime is
red symbol, and nighttime is blue one.

Fig. 9 Same as Fig. 8 except for corrected POM-02 precipitable water vapor.

Fig. 10 Scatter plot of GPS and Radiosonde precipitable water vapor.

Fig. 11 Scatter plot of PWV from GPS and corrected PWV from POM-02. (a) daytime
(red), (b) nighttime (blue), (c) overlapping daytime (red) with nighttime (blue).

Fig. 12 Examples of time series of GPS (red), POM-02 daytime (green) and
nighttime (blue) corrected precipitable water vapor.





Table 1 Examples of calibration coefficient $V_{S0}$ for the solar measurement

| Wavelength (nm) | 340 | 380 | 400 | 500 | 675 | 870 | 940 | 1020 |
|---|---|---|---|---|---|---|---|---|
| $V_{S0}$ ($\times 10^{-4}$) | 0.1799 | 0.1882 | 1.603 | 3.174 | 3.444 | 2.299 | 1.055 | 1.077 |

| Wavelength (nm) | 1225 | 1627 | 2200 |
|---|---|---|---|
| $V_{S0}$ ($\times 10^{-4}$) | 0.9305 | 1.321 | 0.7873 |

Table 2 Measurement range before (current) and after modification (new) of POM-02.
In and In−1 are the upper and lower limit current (unit: A), respectively.

| | | Current | | New | |
|---|---|---|---|---|---|
| 1 | $I_1 - I_2$ | $2.5\times10^{-3}$ – $2.5\times10^{-4}$ | | $2.5\times10^{-3}$ – $1.25\times10^{-4}$ | |
| 2 | $I_2 - I_3$ | $2.5\times10^{-4}$ – $2.5\times10^{-5}$ | | $1.25\times10^{-4}$ – $6.25\times10^{-6}$ | |
| 3 | $I_3 - I_4$ | $2.5\times10^{-5}$ – $2.5\times10^{-6}$ | | $6.25\times10^{-6}$ – $3.125\times10^{-7}$ | |
| 4 | $I_4 - I_5$ | $2.5\times10^{-6}$ – $2.5\times10^{-7}$ | | $3.125\times10^{-7}$ – $1.5625\times10^{-8}$ | |
| 5 | $I_5 - I_6$ | $2.5\times10^{-7}$ – $2.5\times10^{-8}$ | | $1.5625\times10^{-8}$ – $7.8125\times10^{-10}$ | |
| 6 | $I_6 - I_7$ | $2.5\times10^{-8}$ – $2.5\times10^{-9}$ | | $7.8125\times10^{-10}$ – $3.90625\times10^{-11}$ | |
| 7 | $I_7$ | $2.5\times10^{-9}$ – $0.0$ | | $3.90625\times10^{-11}$ – $0.0$ | |
| | $I_n$ | $I_n = I_{n-1}/10$ | | $I_n = I_{n-1}/20$ | |

Table 3 Coefficients of the regression equation for reflectance correction factor C

| Wavelength (nm) | $A_c$ | $B_c$ |
|---|---|---|
| 340 | $1.3404\times10^{-5}$ | 0.98027 |
| 380 | $1.3512\times10^{-5}$ | 1.0674 |
| 400 | $3.0760\times10^{-6}$ | 1.0058 |
| 500 | $2.2487\times10^{-6}$ | 1.1600 |
| 675 | $4.8644\times10^{-6}$ | 1.0840 |
| 870 | $3.4967\times10^{-6}$ | 1.0855 |
| 940 | $7.2405\times10^{-8}$ | 1.1532 |
| 1020 | $6.7912\times10^{-6}$ | 1.0559 |
| 1225 | $9.0288\times10^{-5}$ | 1.0572 |
| 1627 | $2.3828\times10^{-5}$ | 1.0810 |
| 2200 | $3.7545\times10^{-5}$ | 0.95311 |

$C = A_c \cdot g^2 + B_c$ , $g$: phase angle (degrees)



Table 4 Results of comparison between NIES/HSRL and POM-02 aerosol optical depth

| POM-02 | Bias | RMSE | CR | $C_1$ | C.I. of $C_1$ (95%) | $C_2$ | C.I. of $C_2$ (95%) | RMSE of reg. | NO of obs. |
|---|---|---|---|---|---|---|---|---|---|
| Sun+Moon | 0.0437 | 0.0839 | 0.8266 | 0.9611 | ±0.0295 | 0.0486 | ±0.0049 | 0.0715 | 1889 |
| Sun | 0.0432 | 0.0866 | 0.7650 | 0.8877 | ±0.0425 | 0.0573 | ±0.0068 | 0.0743 | 1192 |
| Moon | 0.0466 | 0.0838 | 0.8825 | 1.0477 | ±0.0414 | 0.0405 | ±0.0074 | 0.0694 | 702 |

RMSE: Root mean square error

CR: Correlation coefficient

$C_1$ and $C_2$: coefficients of regression line ($\tau_{HSRL} = C_1 \cdot \tau_{POM-02} + C_2$)

C.I. of $C_1$ (95%): 95% confidential interval of $C_1$

C.I. of $C_2$ (95%): 95% confidential interval of $C_2$

RMSE of reg.: RMSE of regression line

Table 5 Same as Table 4 except for radiosonde and POM-02 precipitable water vapor

| POM-02 | Bias | RMSE | CR | $C_1$ | C.I. of $C_1$ (95%) | $C_2$ | C.I. of $C_2$ (95%) | RMSE of reg. | No. of obs. |
|---|---|---|---|---|---|---|---|---|---|
| Sun+Moon | −0.2477 | 0.3037 | 0.9946 | 0.7948 | ±0.0138 | −0.0057 | ±0.0196 | 0.0658 | 141 |
| Sun | −0.2206 | 0.2764 | 0.9945 | 0.8041 | ±0.0165 | −0.0044 | ±0.0223 | 0.0661 | 104 |
| Moon | −0.3259 | 0.3726 | 0.9966 | 0.7811 | ±0.0214 | −0.0212 | ±0.0343 | 0.0508 | 37 |

PWV, Bias, RMSE, RMSE of reg.: g/cm²

$C_1$ and $C_2$: coefficients of regression line ($PWV_{POM-02} = C_1 \cdot PWV_{Sonde} + C_2$).

Table 6 Same as Table 4 except for radiosonde and corrected POM-02 precipitable water vapor.

| POM-02 | Bias | RMSE | CR | $C_1$ | C.I. of $C_1$ (95%) | $C_2$ | C.I. of $C_2$ (95%) | RMSE of reg. | No. of obs. |
|---|---|---|---|---|---|---|---|---|---|
| Sun+Moon | −0.0027 | 0.0830 | 0.9946 | 1.0042 | ±0.0173 | −0.0077 | ±0.0246 | 0.0829 | 142 |
| Sun | 0.0115 | 0.0848 | 0.9945 | 1.0160 | ±0.0206 | −0.0061 | ±0.0278 | 0.0831 | 105 |
| Moon | −0.0454 | 0.0794 | 0.9966 | 0.9869 | ±0.0271 | −0.0272 | ±0.0434 | 0.0643 | 37 |

$C_1$ and $C_2$: coefficients of regression line ($PWV_{POM-02}(corrected) = C_1 \cdot PWV_{Sonde} + C_2$).



Table 7 Same as Table 4 except for GPS and radiosonde precipitable water vapor.

| Sonde | Bias | RMSE | CR | $C_1$ | C.I. of $C_1$ (95%) | $C_2$ | C.I. of $C_2$ (95%) | RMSE of reg. | No. of obs. |
|---|---|---|---|---|---|---|---|---|---|
| Sonde | 0.0770 | 0.2229 | 0.9791 | 0.9425 | ±0.0233 | 0.1572 | ±0.0403 | 0.2007 | 274 |

$C_1$ and $C_2$: coefficients of regression line ($PWV_{Sonde} = C_1 \cdot PWV_{GPS} + C_2$).

Table 8 Same as Table 4 except for GPS and corrected POM-02 precipitable water vapor.

| POM-02 | Bias | RMSE | CR | $C_1$ | C.I. of $C_1$ (95%) | $C_2$ | C.I. of $C_2$ (95%) | RMSE of reg. | No. of obs. |
|---|---|---|---|---|---|---|---|---|---|
| Sun+Moon | 0.0159 | 0.2050 | 0.9664 | 0.9032 | ±0.0089 | 0.1255 | ±0.0122 | 0.1896 | 2826 |
| Sun | 0.0072 | 0.1939 | 0.9706 | 0.9056 | ±0.0097 | 0.1164 | ±0.0137 | 0.1787 | 2046 |
| Moon | 0.0391 | 0.2232 | 0.9527 | 0.9132 | ±0.0221 | 0.1292 | ±0.0279 | 0.2106 | 671 |

$C_1$ and $C_2$: coefficients of regression line ($PWV_{POM-02}(corrected) = C_1 \cdot PWV_{GPS} + C_2$).





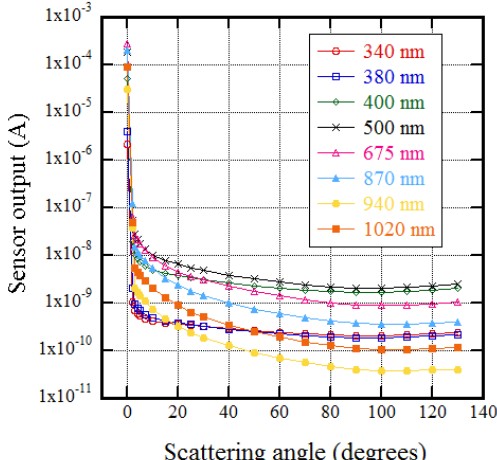

Fig. 1 Examples of sensor output for solar direct irradiances and scattered sky radiances from POM-02.

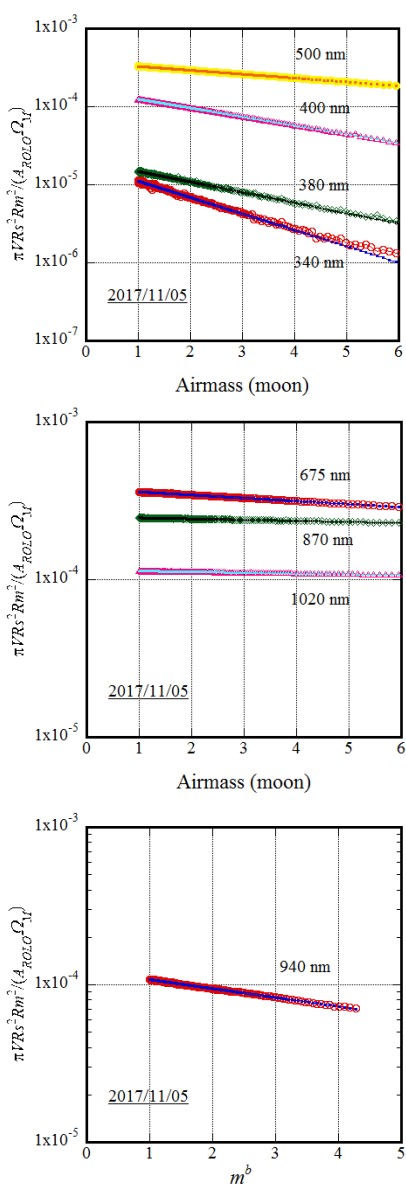

Fig. 2 Examples of a Langley plot in the visible and near-infrared region.





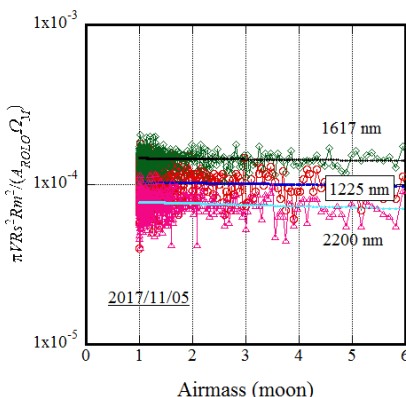

Fig. 3 Examples of Langley plots in the shortwave infrared region.

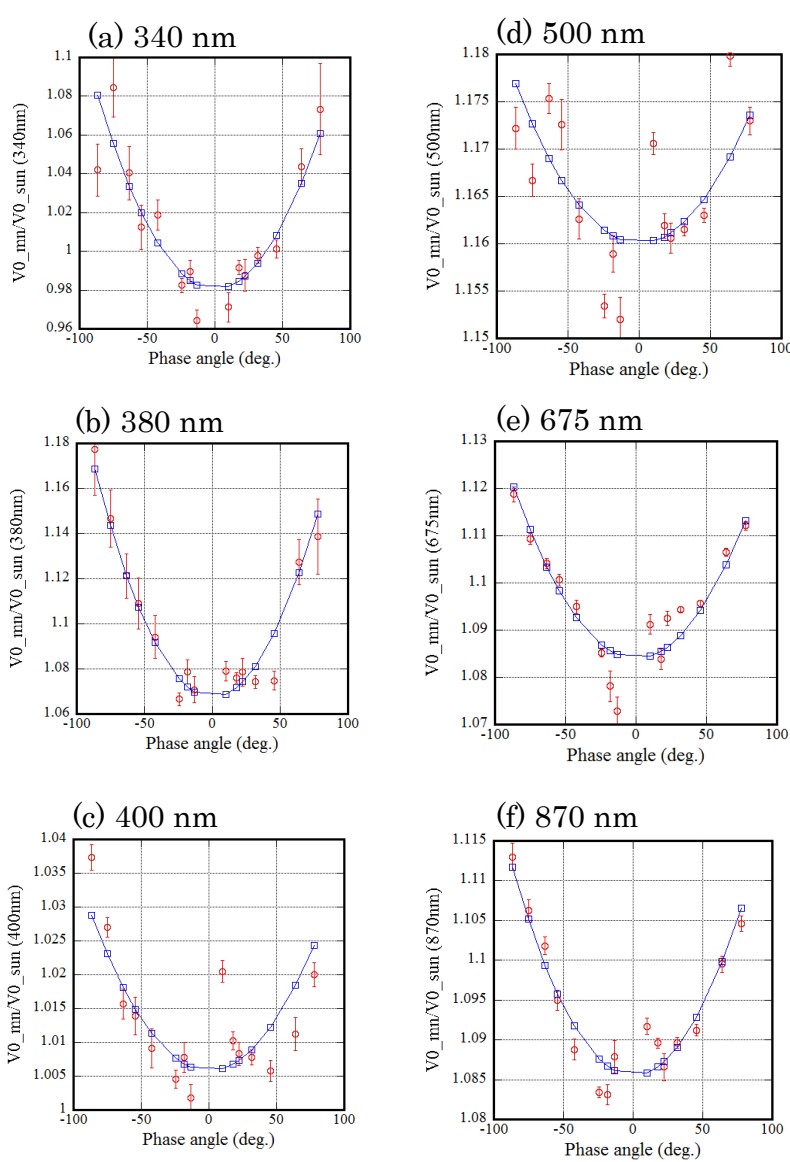

Fig. 4 to be continued.

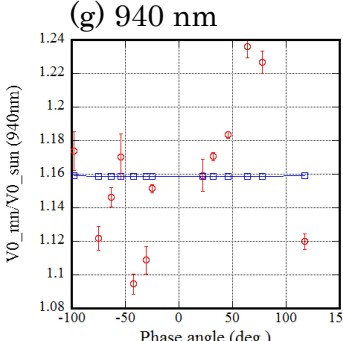

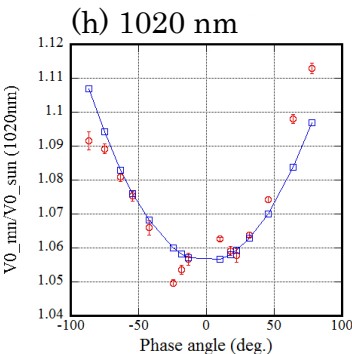

Fig. 4 Relationship between phase angle and reflectance correction factor $C = V_{m0} / V_{S0}$ in visible

and near-infrared region. A regression curve ($C = A_c \cdot g^2 + B_c$, $g$ : Phase angle) was also plotted.

to be continued.



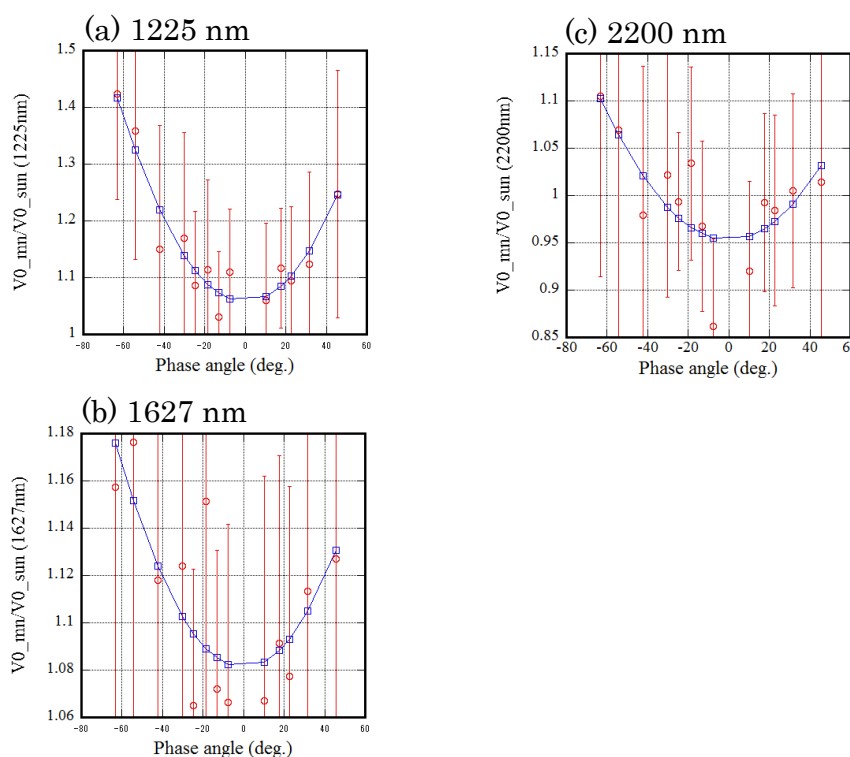

Fig. 5 Relationship between phase angle and reflectance correction factor $C = V_{m0} / V_{S0}$ in shortwave infrared region. A regression curve ($C = A_c \cdot g^2 + B_c$, $g$ : Phase angle) was also plotted.

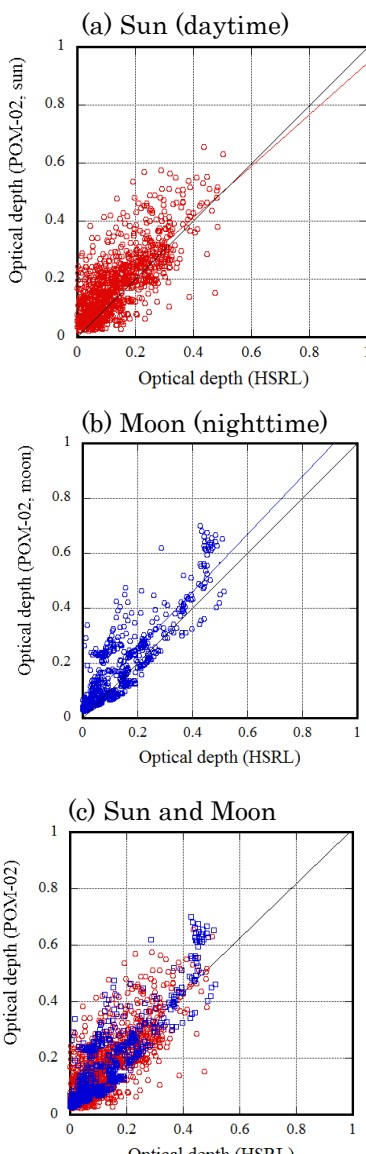

Fig. 6 Scatter plot of NIES/HSRL and POM-02 aerosol optical depth. (a) Daytime (red),
(b) nighttime (blue), (c) overlapping daytime (red) with nighttime (blue).

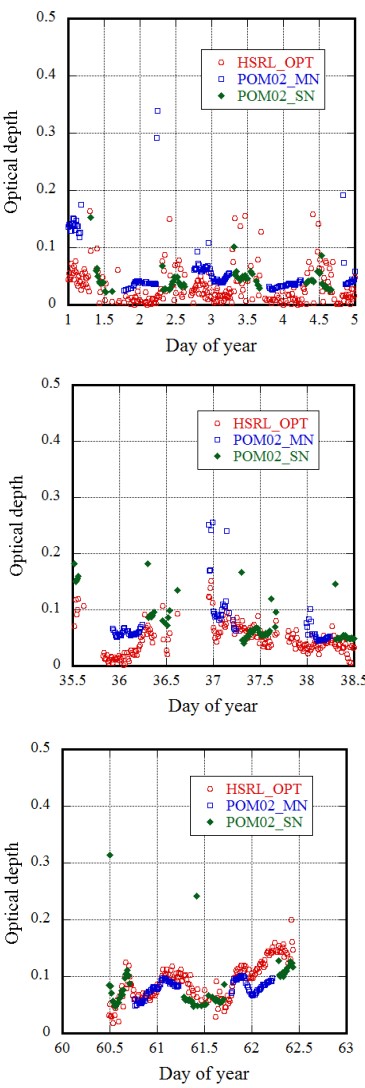

Fig. 7 Examples of time series of NIES/HSRL (red), POM-02 daytime (green), and nighttime (blue) aerosol optical depths.





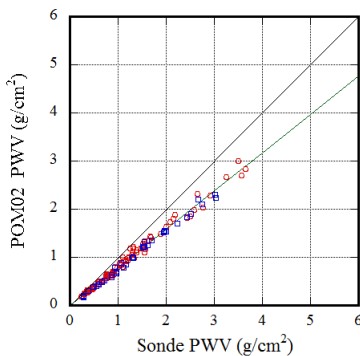

Fig. 8 Scatter plot of radiosonde and POM-02 precipitable water vapor. Daytime (nighttime) observations are indicated by red (blue) symbols.

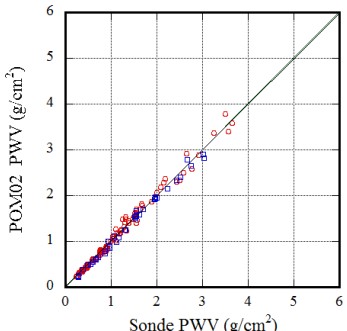

Fig. 9 Same as Fig. 8 except for corrected POM-02 precipitable water vapor.

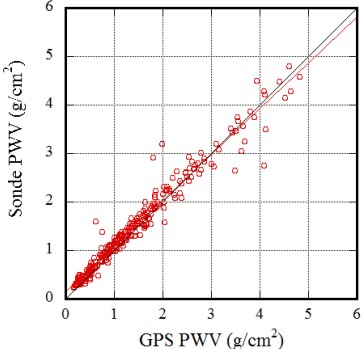

Fig. 10 Scatter plot of GPS and radiosonde precipitable water vapor.



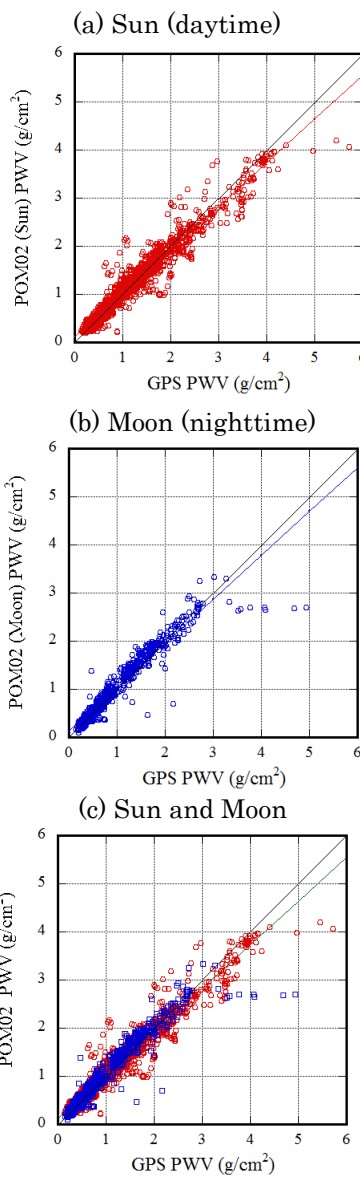

Fig. 11 Scatter plot of PWV from GPS and corrected PWV from POM-02. (a) Daytime (red), (b) nighttime (blue), (c) overlapping daytime (red) with nighttime (blue).



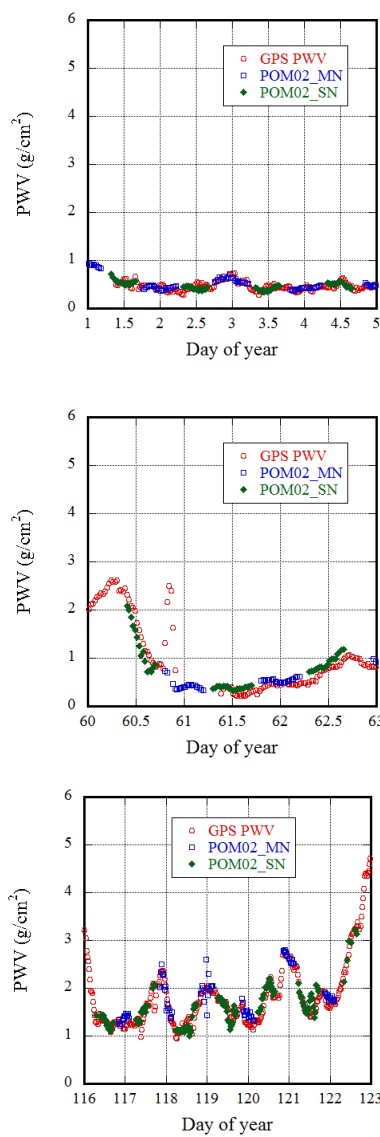

Fig. 12 Examples of time series of GPS (red), POM-02 daytime (green), and nighttime (blue) corrected precipitable water vapor.