# Peer review of "Title: Nocturnal aerosol optical depth measurements with modified skyradiometer"

_Atmospheric Measurement Techniques, 2019_

## Referee Comment (RC1) · Anonymous Referee #1 · 11 Jul 2019

The work presented in this paper is valuable as indicates that the lunar photometry technique is gaining space in the scientific community and arising interest from the companies producing the needed equipment.

General comments:

I have doubts on using AOD obtained from LIDAR as the truth. Usually is the opposite, you use AOD from photometer to constrain the LIDAR results. This is also in some way recognized by you at L474-477.

I suggest to read and eventually cite also the last paper from Barreto et al. 2019 about intercomparison between CIMEL and PFR lunar versions.

[Figure]

https://doi.org/10.1016/j.atmosenv.2019.01.006

I think sections 2.2 and 2.3 could be merged. There are repetitions between them. I don't understand the last part of the last sentence of section 2.2: "depending on the aerosol optical depth....".

Did you wrote an implementation of the ROLO equation? Did you used filter functions of just central wavelengths?

Eq4 In my opinion is not necessary to define C1 and C2, it's confusing to me.

Eq7 Same as for Eq4. By the way, you are using the same symbols C1 and C2

Finally, I suggest to ask a mother-tongue people to check the manuscript.

Specific comments:

L34 I suggest "reflectance estimated by the Robotic..." L34 Maybe a reference to the ROLO paper is required here L38 "visible and near-infrared" L38 "This indicates..." or "could indicate" L65 AERONET L131 Remove "Prede Co Ltd., Japan". Is a repetition L157 Could you please explain this sentence? Where you get 2x10-5? L177 "Therefore, it is difficult..." L220 The equation is reported without any introduction words. L305-308 Repetition. Please remove L373-374 Even if the meaning is clear to me, I suggest to rephrase the sentence in a clearer way. L394-395 Remove the repetition of "the detector output" -> "and hence may be..." L406-409 Please rephrase the sentence in a clearer way. L413 The sentence about the calibration factors at 940 nm is not clear to me. L418 Please check the "C(=" L448-451 This sentence is a repetition. L491 UTC, not UCT L494-496 You repeated many times in the text this explanation about the statistics shown in the tables. I think one time is enough. L602-603 The sentence about the ROLO model is not clear to me. Please rephrase

Table1 Which are the units? Amperes?

Fig2, Fig3 Maybe you could improve/simplify the labels of the y axes

[Figure]

---

## Referee Comment (RC2) · Anonymous Referee #2 · 23 Jul 2019

**General comments:**

Overall, this is a well-written paper describing the modification of the current PREDE-POM radiometers to extend the aerosol monitoring at night-time using the moon as the light source. Considering this to be the reference instrument of one of the most important ground-based networks such as Skynet, the adoption of this new system will provide valuable information for atmospheric research and will certainly be widely used. I consider that this manuscript fits perfectly into the scope of AMT. I recommend publishing the manuscript, but there are some important discussion points and details that I would like the authors to address before its final publication:

- This referee is concerned about the lack of continuous lunar measurements shown in this paper. The authors measured 2 moon cycles in MLO but they do not show any results in terms of AOD. Furthermore, they have 6 moon cycles in Tsukuba, but only a few days are shown in figure 7. They claim they can perform measurements at higher phase angles, but there is no evidence of this in the text. Figure 7 offers no information as to what phase angle corresponds with these measurements, and they do not constitute proof of continuous lunar measurements in themselves (see also specific comments).

- Moreover, the choice of the NIES/HSRL as the only validation analysis of this paper seems to be an important weak point (it needs interpolation or the assumption of a constant extinction coefficient at altitudes of less than 500m). There is a Cimel CE318-T taking measurements at MLO during the period the Prede was calibrating there. I suggest the authors include this interesting comparison in their study. Furthermore, the statistics involved in the day/night/day continuity of AOD will give important information about the instrument's performance.

- In the manuscript there is sometimes a lack of required scientific and analytical rigor and therefore, I suggest that potential subjective sentences be avoided. I would like to highlight the following examples:

    - The ability of POM-02 to perform measurements beyond the quarters. In the conclusions the authors stated that this instrument is able to measure up to 120 degrees in phase angle. This value is quite surprising because it is the first time I have read it in the text. Notwithstanding, considering this to be the limit of your instrument's measuring capabilities, the authors must provide proof of that. Sections 2 and 3 describe a sensor able to perform measurements between ±90 degrees and the authors claim that it is possible to extend this range if the instrument is accurately installed. Firstly, this assumption is vague and imprecise, and secondly there is no

evidence in the manuscript showing this instrument is capable of providing measurements in this claimed phase angle range.

- The authors use only the ROLO model in terms of reflectance, assuming the relative change in the model's reflectance is correct. But as Kieffer and Stone (2005) stated, direct dependence on solar model will cancel itself out as long as the same model is used in going from irradiance to reflectance and back again. That means, the authors are introducing an "uncounted" error in this step, and they are not correcting their model implementation by using the Apollo spectra either. The authors should give some discussion about this effect, taking into account that they attribute the residuals of the C coefficient solely to errors of the ROLO model itself or to lunar librations. There are also other sources of error such as interpolation/extrapolation of the ROLO coefficients, temperature correction, Langley fitting, possible non-linearity of PREDE-POM sensor, noise, among others.

- Similar precision and accuracy of night-time measurements in comparison to daytime: I suggest the authors present some evidence which justifies this statement or avoid giving vague assumptions. I understand that an uncertainty analysis could be out of the scope of this paper (although it would be very illuminating). However, without this study, the authors should not state that the uncertainty at daytime is similar to the one at night-time.

**Specific comments:**

This is not the first time that an attempt for ROLO correction has been published. Therefore, I recommend including some discussion about similar corrections already presented in previous publications.

Amplification in the new PREDE-POM: I am not sure by reading the text if the new instrument includes amplification or not. I read on page 3, lines 170-172, that it can use amplifiers in the visible and nIR spectral range (1 to 7). However, in Eq. 2, there is

no amplification used between sun and lunar measurements. Could you please clarify this?

This is a technical paper aiming to extend AOD capabilities to night-time. Nonetheless, the authors presented a correction of the lunar irradiance model without showing its impact in terms of AOD. Furthermore, this goal seems very ambitious, taking into account that the continuous operation of this instrument has not been appropriately verified.

Page 13, lines 404-417: How many points do you use (C coefficients) to perform your ROLO correction? What about the residuals? Do you use the residuals to estimate the precision of this correction by 1%? Please clarify.

Page 15, lines 469-472: there are references to "similar time variations" or "systematic differences" without quantifying them. The authors also claimed that this figure constitutes a continuous series, but it contains only 11 different nights from 3 different moon cycles (with 4, 4 and 3 nights, respectively). You do not provide any more data for an entire moon cycle, and is this because of clouds, technical problems... Please clarify.

I feel lost with units in Section 2.1. You write Amps almost everywhere, but sometimes I think you are talking about physical units (irradiance or radiance units). Figure 1 presents solar direct and scattered irradiance in Amps. Why do you convert everything into current intensity?

**Technical comments:**

Page 1, line 33: Please define MLO.

Page 1, line 33: I read in the text (page 8 line 258) that the calibration period is September-November. Please verify.

Page 2, line 44: Please define MRI/JMA.

Page 2, line 44: Please define NIES.

Page 2, line 50: As written in general comments, the authors should avoid making

vague assumptions.

Page 2, line 65: there is a typo in AERONET.

Page 3, lines 79-84: Please homogenize if you write acronyms or not.

Page 3, line 98: "in many cases" Please be more specific. Are the authors talking about elastic lidars?

Page 5, section 2.2: I feel the FOV of the instrument is missing here. I assume it is the same as the non-modified Prede, but I consider it will be useful for the reader to include this information.

Page 8, line 257: How many measurements?

Page 9, section 5.1: I recommend the authors to separate this section into two, one for daytime calculations and another one for the methodology applied to night-time. I consider it will certainly improve the readability of this section.

Page 9, line 370: Is the Rm distance also expressed in AU? ROLO refers the reference to the mean Earth-Moon distance in km (384400km).

Page 10, line 325: Why the authors say "it is often used"? Because they use it sometimes and sometimes not? Please clarify.

Page 10, line 351: Please be clearer about "several atmospheric models".

Page 13, line 407: "...the moon and the sun".

Page 19, line 639: This is the first time I have read 120 degrees in the text.

Figure 2: Please label each figure and include some information about each sub-plot. According to page 12, line 392 and Eq. 10, the y-axis of this plot must be $\ln(\pi \cdot V \cdot R_s^2 \cdot R_m^2 / A \cdot \Omega)$?

Figures 6 and 7: Please include information about wavelength.

---

## Short Comment (SC1) · 24 Jul 2019

The article "Nocturnal aerosol optical depth measurements with modified skyradiometer POM-02 using the moon as a light source" presents aerosol optical depth (AOD) measurements acquired at night using the Prede POM-02 radiometer. The exo-atmospheric lunar irradiances used in this work for AOD retrievals are obtained from an implementation of the USGS ROLO lunar model [Kieffer and Stone, 2005 (hereafter K&S)]. The approach presented in this paper utilizes the direct outputs of the ROLO lunar disk reflectance model, K&S Eq. 10, rather than using the model to generate lunar irradiance as it is intended. These direct outputs of K&S Eq. 10 are then interpolated

to the wavelengths of the POM-02 instrument spectral bands. There are significant problems with this approach, and it is not recommended to use the ROLO model in this way.

The lunar disk reflectance spectrum produced from computing K&S Eq. 10 exhibits a large amount of spectral structure, as can be seen in the example of Figure 1 of this comment. This structure is the result of the development process for the ROLO model, where the model coefficients were determined for each of the ROLO bands independently. However, the actual reflectance spectrum of the Moon is known to be smooth, with only broad, shallow absorption features [e.g. Pieters and Mustard, 1988; McCord et al., 1981]. To remove the artificial spectral structure produced by K&S Eq. 10, the USGS ROLO system fits these outputs with a reference lunar reflectance spectrum. This reference spectrum was developed from laboratory spectra of returned Apollo samples, and it is considered representative of the lunar disk as a whole in terms of spectral content (not absolute reflectance). The fitting process smoothes the results of Eq. 10 to give a realistic lunar reflectance spectrum. Figure 1 shows an example of the outcome of this process. It should be apparent from this figure that interpolating between the direct results for the ROLO bands (square symbols) generally will not produce values that align with the fitted spectrum (solid line).

The recommended usage of the ROLO model is to fit the reference reflectance spectrum to the outputs of Eq. 10, where these have been computed for the particular phase angle and librations of the instrument's Moon observation, then convolve this fitted spectrum with the sensor band spectral response and the solar spectral irradiance to give lunar irradiance, as:

$$E_\mathsf{M} = \frac{\Omega_\mathsf{M}}{\pi} \frac{\int A_\mathsf{fit}(\lambda)\, E_\mathsf{Sun}(\lambda)\, S(\lambda)\, d\lambda}{\int S(\lambda)\, d\lambda} \tag{1}$$

where $E_\mathsf{M}$ is the lunar spectral irradiance, $\Omega_\mathsf{M}$ is the solid angle of the Moon, $A_\mathsf{fit}(\lambda)$ is the fitted lunar reflectance spectrum, $E_\mathsf{Sun}(\lambda)$ is the solar spectral irradiance, and $S(\lambda)$ is the sensor band spectral response. The result of Equation 1 must then be corrected for the actual Sun-Moon and Moon-sensor distances of the instrument's Moon observation. The solar spectral irradiance used in this step should be the PMOD-WRC model published by C. Wehrli (`ftp://ftp.pmodwrc.ch/pub/publications/pmod615.asc`). This is because the Wehrli solar model was used to convert the original ROLO irradiance data to reflectance for developing the disk reflectance model, and the same solar model must be used to convert reflectance back to irradiance (Equation 1) to avoid errors caused by differences in the solar spectral models.

Kieffer, H. H. and T. C. Stone (2005), The Spectral Irradiance of the Moon, *Astronom. J.* **129**, 2887–2901

Pieters, C. M. and J. F. Mustard (1988), Exploration of Crustal/Mantle Material for the Earth and Moon Using Reflectance Spectroscopy, *Rem. Sens. Environ.* **24**, 151–178

McCord, T. B. et al. (1981) Moon: Near-Infrared Spectral Reflectance, a First Good Look, *J. Geophys. Res.* **86**, 10,833–10,892

[Figure]

[Figure]

**Fig. 1.** Example of ROLO lunar disk reflectance model output and fitted reference lunar reflectance spectrum

[Figure]

---

## Author Comment (AC3) · 16 Oct 2019

Please see supplement pdf file.

Please also note the supplement to this comment:
https://www.atmos-meas-tech-discuss.net/amt-2019-230/amt-2019-230-AC3-supplement.pdf

---

## Author Comment (AC4) · 16 Oct 2019

We also send main tex as pdf file as supplement.

Please also note the supplement to this comment:
https://www.atmos-meas-tech-discuss.net/amt-2019-230/amt-2019-230-AC4-supplement.pdf

---

## Author Comment (AC5) · 16 Oct 2019

[revised manuscript text omitted]

The Cimel sun photometer used in AERONET has been modified for lunar observation and the aerosol optical depth at night can be estimated (Berkoff et al. 2011; Barreto et al. 2013, 2016, 2017). In addition, a lunar photometer—the Moon Precision Filter Radiometer, LunarPFR (Kouremeti et al. 2016)—has been developed by the Physical Meteorological Observatory in Davos (PMOD), which serves as the World Radiation Center (WRC), based on the sun-PFR experience. Using these instruments and stellar photometers, a multi-instrument nocturnal intercomparison campaign was conducted to evaluate nighttime aerosol measurements and lunar irradiance models (Barreto et al. 2019).

In SKYNET, the radiometers POM-01 and POM-02, manufactured by Prede Co. Ltd., Japan, are used. These radiometers are called 'sky radiometers', and measure both the solar direct irradiance and sky-radiances (Takamura et al. 2004). The sky radiometers POM-01 and POM-02 can measure solar direct irradiance and sky-radiances during the daytime and the measured data are used for estimating aerosol characteristics during the daytime (Takamura et al. 2004). In this study, we will aim to measure the optical depth of aerosol using the moon as a light source by modifying POM-02.

In section 2, we describe our modification of the instrument. In section 3, the

ROLO model is briefly explained. In section 4, we briefly describe the data used in this study. In section 5, the calibration method and corresponding results are described. In section 6, we show the results of comparing the aerosol optical depth and precipitable water vapor obtained by continuous observation with those obtained by other independent instruments. We also show the results of comparing the aerosol optical depth and precipitable water vapor before and after sunrise and sunset using the continuous observation data. Furthermore, we show the results of a comparison between the AERONET and POM-02 data during the period of the MLO calibration measurement.

2.   Modification of instrument

In the modification of the POM-02 for solar observation, only the amplifier and the position sensor were changed. The other components, e.g., detectors, filters, and lenses, are not changed. Therefore, the magnitude of the solid view angle (field of view) for the new POM-02 is the same as in the non-modified POM-02.

Measurements can still be obtained in the daytime using the modified POM-02.

2.1 Adjustment of Amplifier

The sky radiometer POM-02 is designed to measure the direct solar irradiance and the scattered sky radiance with a single radiometer. An example of the calibration constant, which is the sensor output for the extra-terrestrial solar irradiance at the mean earth-sun distance (1 astronomical unit (AU)) at the reference temperature, is shown in Table 1. The calibration constant is $1.8 \times 10^{-5}$ to $3.4 \times 10^{-4}$ A in the visible and near-infrared region, and $7.9 \times 10^{-5}$ to $1.3 \times 10^{-4}$ A in the short-wavelength infrared region. Figure 1 shows an example of measurements of scattered radiances in the visible and near-infrared wavelength region. The output for the scattered radiance from the sky is $1 \times 10^{-7}$ to $1 \times 10^{-10}$ A, and this value is $1 \times 10^{-6}$ smaller than the output for the direct solar irradiance. The direct lunar irradiance is $1 \times 10^{-5}$ as strong as the direct solar irradiance during a full moon, and $1 \times 10^{-6}$ during a half-moon (Berkoff et al. 2011). From Table 1, the calibration constants at 340 and 380 nm are $1.8 \times 10^{-5}$ and

$1.9 \times 10^{-5}$ (about $2 \times 10^{-5}$), respectively. Therefore, the output for the direct lunar irradiance during the half moon is about $2 \times 10^{-5} \times 10^{-6} = 2 \times 10^{-11}$ in the 340 and 380

nm channels. This is close to the detectable limits of the current POM-02. Without modification, it is possible to measure the direct lunar irradiance with the current POM-02 except for wavelengths between 340 and 380 nm where the sensitivity of the detector is low and wavelengths of 1225, 1627, and 2200 nm with poor S/N.

Table 2 shows the measurement ranges before and after modification of POM-02. POM-02 measures input energy in seven ranges according to the magnitude of the input energy, and the measured value is digitized with 15 bits. After modification the measurement ranges are slightly expanded, and the measurement limit depends on the magnitude of the dark current and the magnitude of the noise. The sensor output takes into account the magnification of the amplifier, and the same amplifier was used for both the solar and lunar measurements.

The dark current of the detector in the visible and near-infrared region was about $5 \times 10^{-13}$ A, and the RMS of the random component of the noise was $4 \times 10^{-14}$ A. In consideration of these values, the new POM-02 can use amplifiers for measurement ranges 1 to 7 and the minimum meaningful current is about $4 \times 10^{-13}$ A (~RMS×10) in the visible and near-infrared region. This value is smaller than the output for the direct solar irradiance by a factor of $1 \times 10^{-8}$ to $1 \times 10^{-9}$.

The dark current of the detector in the shortwave infrared wavelength region was about $1.5 \times 10^{-8}$ A, and the RMS of the random component of the noise was $4 \times 10^{-11}$ A. The measurement range is limited due to the large dark current. The new POM-02 can use amplifiers for measurement ranges 1 to 5 and the minimum meaningful current is about $4 \times 10^{-10}$ A (~RMS×10). This value and the magnitude of the measured value of the direct lunar irradiance are comparable. Therefore, it is difficult to measure the direct lunar irradiance even with the new POM-02 in the shortwave infrared wavelength region.

2.2 Sun and moon position sensor

The tracking of the sun and the moon is based on the calculated position. The moon positions are calculated with the simplified formula in Nagasawa (1981). The necessary software is installed in the firmware of POM-02. Deviations may occur even if the instrument is pointed in the calculated direction due to errors in the moon position calculation, instrument installation errors, misalignment of the rotation axis, and so on. A position sensor is used to correct this deviation.

A position sensor with a four-quadrant detector is used to adjust the tracking of the sun and the moon. In order to adjust the tracking of the moon, a position sensor incorporating a new electronic circuit to amplify the signal and new software to process the signal data were developed. The new position sensor can be used to track both the sun and the moon.

When the input energy to the position sensor is small, it is difficult to adjust the tracking with the position sensor. The magnitude of the input energy to the position sensor varies depending on the lunar phase and the aerosol optical depth. It was confirmed that the function of the moon tracking adjustment works during the period of the full moon ± about 90 degrees of the phase angle (half-moon).

Whether the position sensor can be used can be determined by a user-specified threshold value. That is, the position sensor can be deactivated when the input energy to the position sensor becomes less than the threshold value. For phase angles larger than the half-moon, the signal of the position sensor was small, and the position sensor was deactivated.

When the position sensor is not functioning, tracking is performed based on the calculated values. When comparing the moon position calculated by this simplified formula with that calculated using the NASA SPICE toolkit (Acton 1996), the difference in the zenith angle is less than 0.01 degrees, and the difference in the azimuth angle is less than 0.04 degrees. The center of the field of view has a flat region of ±0.5 degrees; the flat region is ±0.25 degrees in the solar disk scan. The apparent diameters of the sun and the moon are about 0.5 degrees. Since the calculation error of the moon position is less than 0.25 degrees, if the misalignment of the rotation axis is negligible and POM-02 is installed correctly, it is possible to track the moon using only the calculated positions. In fact, measurements could be made on the day of a full moon ± 10 days (phase angle about 120 degrees). Figure 2

shows an example of the measurements on Oct. 14, 2017 at NOAA/MLO. In this example, the phase angle of the moon is from 117.6 to 118.0 degrees.

3. Robotic Lunar Observatory (ROLO) irradiance model

[revised manuscript text omitted]

normalized by 384,400 km (the mean radius of the moon's orbit around the earth).

$\tilde{A}_{ROLO}$ is the smoothed ROLO reflectance adjusted to the laboratory reflectance spectra of the Apollo 16 samples. $\tilde{A}_{ROLO}$ is calculated using the lunar reflectance

$A_{ROLO}$ with the ROLO irradiance model by the method shown in Kieffer and Stone (2005) (see Appendix A).

Let $\tilde{A}_{ROLO} = F_C A_{ROLO}$ , where $F_C$ is a constant for smoothing (see Appendix A).

Using this equation, eq. (8) becomes

$$V(\lambda_0) = \frac{F_C A_{ROLO}}{\pi} \Omega_M \frac{V_{S0}(\lambda_0)}{R_S^{\ 2}} \cdot \frac{1}{R_m^{\ 2}} \exp(-m(\theta)\tau(\lambda_0))\bar{T}_{gas}(\lambda_0,\theta) \tag{9}$$

It is known that the aerosol optical depth retrieved using the ROLO reflectance
contains an error, which is dependent on the phase angle (Barreto et al. 2016, 2017,
2019, Jurišek and Prouza 2017). We assume that there is an error in the ROLO
reflectance and that the correct lunar reflectance is proportional to the ROLO
reflectance. This indicates that the relative variation in the ROLO model reflectance
is assumed to be correct. Let the proportional constant be denoted $C'$ , and $A_{ROLO}$
in eq. (9) be replaced with $C' \cdot A_{ROLO}$ . Equation (9) then becomes

$$V(\lambda_0) = \frac{F_C C' A_{ROLO}}{\pi} \Omega_M \frac{V_{S0}(\lambda_0)}{R_S^{\ 2}} \cdot \frac{1}{R_m^{\ 2}} \exp(-m(\theta)\tau(\lambda_0))\bar{T}_{gas}(\lambda_0,\theta)$$
$$= \frac{C A_{ROLO}}{\pi} \Omega_M \frac{V_{S0}(\lambda_0)}{R_S^{\ 2}} \cdot \frac{1}{R_m^{\ 2}} \exp(-m(\theta)\tau(\lambda_0))\bar{T}_{gas}(\lambda_0,\theta) \tag{10}$$

where $F_C C'$ is substituted with $C$ .

In the case of no "gas absorption", taking the logarithm of the equation leads to

$$\ln(\frac{\pi V(\lambda_0)}{A_{ROLO}\Omega_M} R_S^{\ 2} R_m^{\ 2}) = \ln C V_{S0}(\lambda_0) - m(\theta)\tau(\lambda_0)$$
$$= \ln V_{m0}(\lambda_0) - m(\theta)\tau(\lambda_0) \tag{11}$$
$$= C_1'' m(\theta) + C_2''$$

where $V_{m0}(\lambda_0) = C V_{S0}(\lambda_0)$ . $C_2'' = \ln V_{m0}$ is determined from the ordinate intercept of a least-square fit when one plots the left-hand side of the above equation versus
airmass $m(\theta)$ .

$V_{S0}$ can be determined by applying the Langley method to data taken during the
daytime. If $V_{S0}$ is determined, the coefficient $C$ can be determined by taking the
ratio of $V_{m0}$ and $V_{S0}$ . If the coefficient $C$ is 1, the reflectance of the ROLO model
will be correct. If the coefficient $C$ is greater than 1 (less than 1), the reflectance in
the ROLO model is under-estimated (over-estimated).

5.2 Results

Examples of Langley plots in the visible and near-infrared wavelengths are shown in Fig. 3. In these examples, the regression lines can be well determined for any wavelength. $C_2'' = \ln V_{m0}$ is determined from the ordinate intercept of the regression line (see eq. (11)). At the 340 nm wavelength, the regression line tends to deviate from the measured values in the region of airmasses larger than 6. It is presumed that the detector output at the 340 nm wavelength is small and hence may be nonlinear. The output at the time of observation was about $1\times10^{-12}$ A. When using output values less than this, the user needs to treat their results with caution. At the

940 nm wavelength, the modified Langley method was applied. In this example, the regression line provides a good fit.

In Fig. 4, examples of the Langley plot in the shortwave infrared region (1225, 1627,

2200 nm) are shown. The detector output of these channels range from $2\times10^{-10}$ to

$5\times10^{-10}$ A, and the root mean square error of the random noise is $4\times10^{-11}$ A. The ratio of noise to detector output is large and it is difficult to use these channels for estimating the aerosol optical depth.

In Fig. 5, the relationship between the coefficient $C(=V_{m0}/V_{S0})$ and the phase angle in the visible and near-infrared wavelength region (from 340 to 1020 nm) is shown. As shown in the previous section, the corrected lunar reflectance is assumed to be proportional to the ROLO reflectance, and the proportional coefficient $C$ is the ratio of the calibration constant for the moon and the sun. That is, the coefficient $C$

indicates the error of the ROLO reflectance, and thus more accurate reflectance can be obtained by multiplying the ROLO reflectance by the coefficient $C$. As can be seen from this figure, the coefficient $C$ is often greater than 1 and depends on the phase angle. At most wavelengths, the coefficient $C$ is small when the absolute value of the phase angle is small (near the full moon) and increases as the absolute value of the phase angle increases. The range of $C$ is 0.95 to 1.18. The absorption band of water vapor is at the 940 nm wavelength. Water vapor in the atmosphere tends to fluctuate. Therefore, it is difficult to make accurate Langley plots, and the accuracy of both $V_{S0}$ and $V_{m0}$ is poor. Therefore, no clear relationship between $C$

and the phase angle is found, but the coefficient $C$ is about 1.16. The fact that $C$ is larger than 1 means that the reflectance of the ROLO irradiance model is underestimated.

In Fig. 6, the relationship between the coefficient $C(=V_{m0}/V_{S0})$ and the phase angle in the shortwave infrared wavelength region (1225, 1627, 2200 nm) is shown.

In these channels, the error for $C$ is large, but the coefficient $C$ depends on the
phase angle as in the visible and near-infrared wavelength region; $C$ is small when
the phase angle is near zero and increases as the absolute value of the phase angle
increases.

In this study, the phase angle dependence of the coefficient $C$ is approximated by
a quadratic equation of the absolute value of the phase angle:

$$C = A_c \cdot g^2 + B_c \qquad\qquad (12)$$

where $g$ is the phase angle.

That is,

$$V_{m0} = V_{S0} \cdot (A_c \cdot g^2 + B_c) \qquad\qquad (13)$$

The coefficients $A_c$ and $B_c$ are shown in Table 3. The regression line was plotted
in Figs. 5 and 6. By using this approximation, the reflectance of the ROLO model can
be estimated to within 1% in most channels. By using this approximation, the data
processing to estimate the aerosol optical depth from the measured value becomes straightforward. The coefficients, $F_C$, for smoothing the ROLO reflectance are also shown in Table 3. The coefficients $A_c'$ and $B_c'$ of the regression equation when using the smoothed ROLO reflectance are also given.

The size of the error in the reflectance in the ROLO irradiance model is dependent
on the phase angle. The ROLO reflectance was obtained by dividing the lunar
irradiance measured by Kieffer and Stone (2005) by the solar spectral irradiance of
the 1985 Wehrli Standard Extraterrestrial Solar Irradiance Spectrum (Wehrli 1985,
Neckel and Labs 1981). The solar spectral irradiances are dependent on the solar
spectral models. Therefore, the ROLO reflectance includes an error due to the error
in the solar spectral irradiance of 1985 Wehrli. Instrument performance, data
processing, and so on are also sources of error. In this study, $C$ is approximated as a
symmetric quadratic equation of the phase angle, but the phase angle dependence of
$C$ is asymmetric (see Figs. 5 and 6). The applicable range of the ROLO reflectance
model is a phase angle of about 95 degrees or less. In order to improve the accuracy of
the ROLO reflectance model and expand its application range, it is necessary to
further accumulate the reflectance data of the moon.

6. Results of comparison

In order to validate the estimations of AOD and PWV, we compared them with the
AOD and PWV obtained by independent methods. We investigated whether there is a difference between daytime and nighttime measurements, and compared the
measurements for the daytime and nighttime with measurement data which was
recorded independently of POM-02 and has the same accuracy and precision in the
daytime and nighttime.

Furthermore, the continuity of the AOD and PWV before and after sunrise and
sunset was investigated, and the AOD and PWV of AERONET and POM-02 at MLO
were also compared.

[revised manuscript text omitted]

In Fig. 13, only limited examples were shown, but in the Supplement, the time series of the PWV at Tsukuba for 5 months is shown in Fig. S2. In addition, the time series of the comparison between GPS and POM-02 PWV for 5 months is shown in

Fig. S4.

6.3 Comparison of AOD (PVW) before and after sunrise and sunset

The comparison of the AOD (PWV) before and after sunrise and sunset is used to evaluate the moon photometry (Berkoff et al. 2011, Barreto et al. 2013, 2016, 2017,

2019).

Before and after sunrise (sunset), the AOD before sunrise (after sunset) is the average of the data with a solar altitude angle between −10 and −15 degrees, with a lunar phase angle less than 100 degrees, and with a lunar altitude angle of more than 10 degrees. The AOD after sunrise (before sunset) is the average of the data with a solar altitude angle between 10 and 15 degrees. Since this comparison is effective when the atmosphere is stable, only data with small variations were
selected; standard deviation / average value is less than 0.1 or standard deviation is
less than 0.02.
Figure 14 shows a scatter plot of the AOD at the wavelengths of 340, 380, 400, 500,
675, 870, and 1020 nm, and the PWV from the 940 nm channel. Table 9 shows the
results of the comparison between the AOD (PWV) from the sun and from the moon
(The contents of Table 9 are the same as Table 4 except for the AOD (PWV) from the
sun and the moon).
The biases at wavelengths of 340 and 380 nm are relatively large, 0.05 and 0.03,
respectively, but the biases at other wavelengths are 0.007 or less. The bias and
RMSE of the PWV are 0.02 and 0.14, respectively, which are comparable to those
from the comparison with POM-02 and the radiosonde or GPS. The correlation
coefficient is high for all wavelengths; 0.65 at a wavelength of 340 nm, and 0.97 or
higher at other wavelengths. Furthermore, the 95% confidence interval of the slope
value of the regression line includes 1, and the 95% confidence interval of the
intercept value includes 0. That is, the regression line is not different from a straight
line with a slope of 1 and zero intercept at the 95% confidence level. From these facts,
the AOD and PWV retrieved using the moon as the light source are considered to be
the same as those retrieved using the sun as the light source at the 95% confidence
level.
6.4 Comparison between AERONET and POM-02
There is an AERONET observation site at MLO. In the nighttime, the AODs at
wavelengths of 500, 675, 870, and 1020 nm, and the PWV can be compared. In
addition to these channels, the AOD at wavelengths of 340, 380, 1020, and 1627 nm
can be compared in the daytime. The AERONET data used here are "level 2.0" in the
daytime and "level 1.5" in the nighttime. There were no "level 2.0" nighttime data.
AERONET "level 1.5" is cloud-screened data but may not have had the final
calibration applied. Thus, these data are not quality assured. AERONET "level 2.0"
has pre- and post-field calibration applied, cloud-screened, and quality-assured data
(see the AERONET homepage, https://aeronet.gsfc.nasa.gov/). The nighttime
comparison in this paper uses the AERONET data without quality assurance.
Figure 15 shows a scatter plot of the AERONET and POM-02 AOD (PWV). The
blue (red) symbols show the daytime (nighttime) data. Both the daytime and the
nighttime data are overlaid; the AOD at wavelengths of 500, 675, 870, 1020, 1627 nm,
and the PWV from the 940 nm channel. The plotted data are the 15-minute averages.

The number of measurements for POM-02 in a 15-minute interval is 10 to 16, and that for AERONET is 1 to 6. Only POM-02 data showing small variations were selected; (standard deviation)/average is less than 0.1 or standard deviation is less than 0.02.

Table 10 shows the results of the comparison between the AERONET and POM-02 aerosol optical depth (precipitable water vapor) (Table 10 is the same as Table 4 except for AERONET and POM-02 aerosol optical depth (precipitable water vapor)). The values at 940 nm are the precipitable water vapor.

In the daytime, from Fig. 15, it can be seen that the differences between AERONET and POM-02 AOD (PWV) are small. The 95% confidence interval for the slope of the regression line does not necessarily include 1, but the slope value is nearly 1: between 0.97 and 1.11. The 95% confidence interval for the intercept of the regression line does not necessarily include 0, but the magnitude of the intercept is 0.01 or less except for the 380 nm channel (0.015). The same can be said for the PWV of the 940 nm channel. In addition, the bias and RMSE are less than 0.01 except for the 380 nm channel (0.015), and those for the PWV at 940 nm are 0.018 and 0.022, respectively. Considering that the accuracy of the calibration constant is 0.5 to 1%, these values seem reasonable. Therefore, it can be inferred that in the daytime, POM-02 can measure the AOD (PWV) with the same level of accuracy as AERONET.

In the nighttime, the atmosphere observed at MLO was pristine, and most of the AOD at 500, 675, 870, and 1020 nm were below 0.02. Considering that the accuracy of the calibration constant is 0.5 to 1%, it is difficult to compare the AOD of AERONET and POM-02. In the nighttime, the slope of the regression line deviates from 1 at several wavelengths, but the bias and the RMSE are less than about 0.01. Therefore, the difference between AERONET and POM-02 is small. The slopes of the regression line for the PWV of 940 nm channel in the daytime and the nighttime are 1.07 and 1.16, respectively. Thus, the daytime and nighttime values differ. In the results of section 6.3, there is almost no difference between the daytime and nighttime values. Therefore, this difference may be due to the lack of quality control in the nighttime data.

7. Summary and conclusion

Aerosol data are often estimated using the solar direct irradiance and the solar scattered radiance. Therefore, the majority of data on aerosol properties are obtained using daytime measurements, and there are few data available on aerosol characteristics at night. In order to estimate the aerosol optical depth (AOD) and the precipitable water vapor (PWV) during the nighttime using the moon as a light source, POM-02 (Prede Ltd., Japan), which is used to estimate aerosol characteristics during the daytime, was modified.

The current POM-02 has the ability to measure the direct irradiance from the moon for some channels in the visible and near-infrared wavelength region without requiring modification. Several modifications were made to also be able to measure the AOD during the nighttime and expand the measurement ranges.

The amplifier was adjusted so that POM-02 could measure up to about $5 \times 10^{-13}$ A: allowing the lunar direct irradiance to be measured in the wavelength range of 340 to 1020 nm.

In order to track the moon based on the calculated value, the simplified formula by Nagasawa (1981) was incorporated into the firmware.

A position sensor with a four-quadrant detector is used to adjust the tracking of the sun and the moon. In order to adjust the tracking of the moon, a position sensor incorporating a new electronic circuit to amplify the signal and new software to process the signal data were developed. The new position sensor can be used to track both the sun and the moon.

The calibration constant was determined by using the Langley method. The measurements of the solar and lunar direct irradiance were conducted at the NOAA/MLO during the period from Sep. 28 to Nov. 7, 2017. Assuming that the correct lunar reflectance is proportional to the ROLO reflectance, the calibration constant for the lunar direct irradiance was determined by using the Langley method. The calibration by the Langley method was successfully performed.

The ratio of the calibration constant for the moon to that for the sun was often greater than 1, where the ratio is a coefficient for correcting the ROLO reflectance and includes a smoothing factor. This ratio shows the error of the ROLO irradiance model. The value of the ratio was 0.95 to 1.18 in the visible and near-infrared wavelength region. This means that the ROLO model often underestimates the reflectance. In addition, this ratio depended on the phase angle: when the phase angle was small (near the full moon), the ratio was small, and as the phase angle became larger, the ratio increased. In this study, this ratio was approximated by the quadratic equation of the phase angle. By using this approximation, the reflectance of the moon can be calculated to within an accuracy of 1% or less.

The continuous measurement of POM-02 was conducted at JMA/MRI from January 2018 to May 2018, and the AOD and PWV were estimated. In order to validate the estimates of the AOD and PWV, we compared them with the AOD and PWV obtained by independent methods. The AOD was compared with the AOD (532

nm) estimated from NIES/HSRL, and the PWV was compared with the PWV from a radiosonde and GPS. In addition, the continuity of the AOD (PVW) before and after sunrise and sunset at Tsukuba was examined, and the AOD (PWV) of AERONET and that of POM-02 at MLO were compared.

Concerning the AOD, there were sometimes systematic differences between NIES/HSRL and POM-02. The cause of the systematic differences seems to be that NIES/HSRL assumes a constant extinction coefficient at altitudes of less than 500 m. The slopes of the linear regression lines during the daytime and nighttime could not be said to be equivalent at the 95% confidence level, but the scatter diagrams of the daytime and nighttime were almost overlapping.

Concerning the PWV, the slopes of the linear regression lines during the daytime and nighttime were equivalent at the 95% confidence level in the comparisons between the PWV from POM-02 and the radiosonde and in the comparison between the PWV from POM-02 and GPS. Furthermore, the scatter diagrams of the daytime and the nighttime data were almost overlapping.

In addition, the comparison of the AOD (PWV) before and after sunrise and sunset showed that the AOD and PWV retrieved using the moon as the light source are the same as those retrieved using the sun as the light source at the 95% confidence level.

The comparison of the AOD (PWV) between AERONET and POM-02 was performed using the data taken during the calibration measurements. The comparison in the daytime showed that POM-02 can measure AOD (PWV) with the same accuracy as AERONET. The comparison in the nighttime showed that the difference in the AOD between AERONET and POM-02 was small. However, since there were a lot of optically thin data and AERONET data are not quality-assured, we cannot make a definite conclusion.

From these facts, the daytime and nighttime AOD (PWV) measurements are statistically almost equivalent. The AODs (PWVs) during the daytime and nighttime for POM-02 are presumed to have the same degree of precision and accuracy within the measurement uncertainty.

The accuracy of the nighttime calibration constant is lower than that for the daytime. The measurement S/N in the nighttime is also worse than that in daytime. Considering these facts, even if there is no statistically significant difference, the magnitude of the error in the AOD (PWV) during the nighttime is not always the same as during the daytime.

In this study, the calibration was performed using about 40 days of data including two full moon days. As a result, it was found that there was an error in the reflectance of the ROLO irradiance model. In the future, it is necessary to accumulate more data for calibration and to reduce the error of the ROLO irradiance model. It is said that the ROLO model can be applied over a phase angle range of about 90 degrees. POM-02 has the ability to measure the direct lunar irradiance up to a phase angle range of about 120 degrees. It is necessary to expand the ROLO

irradiance model so that it can be applied to larger phase angles.

It is now possible to estimate the aerosol optical depth during the nighttime. It is necessary to promote the adoption of this system in the existing observation network.

After that, the data obtained by using this instrument can be used to better understand nighttime aerosol behavior, for the validation of aerosol transport models, and as input data in assimilation systems.

**Appendix A**.

The smoothed ROLO reflectance $\tilde{A}_{ROLO}$ can be obtained by the procedure described in Kieffer and Stone (2005).

The calculated reflectance $A_{ROLO}(g,\Phi,\theta,\phi)$ at the 32 ROLO wavelengths for a specific geometric configuration $(g = 7\deg, \Phi = 7\deg, \theta = 0, \phi = 0)$ is fitted to a composite spectrum of the samples obtained by the Apollo 16 mission with a linear equation of wavelength $\lambda$.

$A_{Apollo} = (a + b\lambda)A_{ROLO}(7,7,0,0)$                    (a1)

where $A_{Apollo}$ is the composite laboratory reflectance spectrum for the Apollo samples of soil (95%) (Apollo 16 sample 62231, (Pieters 1999)) and breccia (5%) (Apollo 16

sample 67455 (Pieters and Mustard 1988)).

The Apollo sample 62231 spectrum is available at http://www.planetary.brown.edu/pds/AP62231.html. The Apollo sample 67455

spectrum is shown in Fig. 8 in the paper of Pieters and Mustard (1988).

The values of the coefficients $a$ and $b$ are not shown in Kieffer and Stone (2005), but were determined here with the least squares method as follows:

$a = 1.640875$

$b = -1.192034 \times 10^{-4}$

where the unit of the wavelengths is nanometers.

By dividing $A_{Apollo}$ by $a + b\lambda$, the smoothed ROLO reflectance for a specific geometric configuration $\tilde{A}_{ROLO}(7,7,0,0)$ can be obtained.

$$\tilde{A}_{ROLO}(7,7,0,0) = A_{Apollo} / (a + b\lambda) \qquad \text{(a2)}$$

The smoothed ROLO reflectance $\tilde{A}_{ROLO}(g,\Phi,\theta,\phi)$ for any viewing geometry is given by the following equation:

$$\begin{aligned}\tilde{A}_{ROLO} &= \frac{\tilde{A}_{ROLO}(7,7,0,0)}{A_{ROLO}(7,7,0,0)} A_{ROLO}(g,\Phi,\theta,\phi) \\ &= F_C A_{ROLO}(g,\Phi,\theta,\phi)\end{aligned} \qquad \text{(a3)}$$

where $F_C = \tilde{A}_{ROLO}(7,7,0,0) / A_{ROLO}(7,7,0,0)$.

The values of $F_C$ are dependent on the interpolation method of the reflectance table and the accuracy of the values read from the figure. The smoothed and adjusted spectrum $\tilde{A}_{ROLO}(7,7,0,0)$ is shown in Fig. A1. The values of $F_C$ determined by the authors are shown in Table A1 and Fig. A2.

**Data availability.**

The data used in this study are available from the corresponding author.

**Author contributions.**

This study was designed by AU, MS, HK, and TM. The measurements for the sky radiometer were conducted by AU, AK, KI, and YW. The adjustment of the amplifier and the development of the position sensor were performed by MS, HK, KI, KK, and

YW. The development of the related software and the data analyses were performed by AU. The manuscript was written by AU, and all authors contributed to editing and revision.

**Competing interests.**

The authors declare that they have no conflict of interest.

**Acknowledgements**

This work was supported by the NIES GOSAT-2 project, Japan. This work was also supported by JSPS KAKENHI Grant Number 17K00531. We would like to thank Dr.

Y. Jin and Dr. T. Nishizawa of NIES for providing the NIES/HSRL data for the comparison of the aerosol optical depth. We also would like to thank Dr. Y. Shoji of

JMA/MRI for providing the GPS data for the comparison of precipitable water vapor. We thank Dr. B. Holben and his staff for their effort in establishing and maintaining the AERONET Mauna Loa site. We would like to thank Dr. T. Stone and two anonymous reviewers for their useful comments.

**References**

[revised manuscript text omitted]

Table 9 Same as Table 4 except for the AOD (PWV) from the sun and the moon.

Table 10 Same as Table 4 except for the AERONET and POM-02 aerosol optical depth (precipitable water vapor).

Figure captions

Fig. 1 An example of sensor output for the solar direct irradiances and the scattered sky radiances by POM-02.

Fig. 2 An example of the measurements taken on Oct. 14, 2017 at NOAA/MLO. The phase angle of the moon (right y-axis) is from 117.6 to 118.0 degrees.

Fig. 3 Examples of the Langley plot in the visible and near-infrared region on Nov. 5, 2017. The y-axis is the equation in parentheses on the left-hand side of eq. (11). (a) 340, 380, 400, 500 nm; (b) 675, 870, 1020 nm; (c) 940 nm, modified Langley method.

Fig. 4 Examples of the Langley plot in the shortwave infrared region.

Fig. 5 Relationship between phase angle and reflectance correction factor $C = V_{m0}/V_{S0}$ in the visible and near-infrared region. A regression curve ($C = A_c \cdot g^2 + B_c$, $g$: phase angle) was also plotted.

Fig. 6 Relationship between phase angle and reflectance correction factor $C = V_{m0}/V_{S0}$ in the shortwave infrared region. A regression curve ($C = A_c \cdot g^2 + B_c$, $g$: Phase angle) was also plotted.

Fig. 7 Scatter plot of HSRL and POM-02 aerosol optical depth at 532 nm. (a) daytime (red), (b) nighttime (blue), (c) overlapping daytime (red) with nighttime (blue).

Fig. 8 Examples of time series of HSRL (red), POM-02 daytime (green) and nighttime (blue) aerosol optical depths at 532 nm. The phase angles (g) during the measurement periods were (a) g = −21.863 to 35.881 degrees, (b) g = 47.454 to 83.190 degrees, and (c) g = −19.150 to 21.573 degrees.

Fig. 9 Scatter plot of radiosonde and POM-02 precipitable water vapor. Daytime (nighttime) measurements are indicated by a red (blue) symbol.

Fig. 10 Same as Fig. 9 except for corrected POM-02 precipitable water vapor.

Fig. 11 Scatter plot of GPS and radiosonde precipitable water vapor.

Fig. 12 Scatter plot of PWV from GPS and corrected PWV from POM-02. (a) daytime (red), (b) nighttime (blue), (c) overlapping daytime (red) with nighttime (blue).

Fig. 13 Examples of time series of GPS (red), POM-02 daytime (green) and nighttime (blue) corrected precipitable water vapor. The phase angles (g) during the measurement periods were (a) g = −21.863 to 35.881 degrees, (b) g = −19.150 to 21.573 degrees, and (c) g = -55.145 to 30.611 degrees.

Fig. 14 Scatter plot of the aerosol optical depth (precipitable water vapor) from the sun and the moon. (a) 340 nm AOD, (b) 380 nm AOD, (c) 400 nm AOD, (d) 500 nm

AOD, (e) 675 nm AOD, (f) 870 nm AOD, (g) 940 nm PWV, (h) 1020 nm AOD.

Fig. 15 Scatter plot of AERONET and POM-02 aerosol optical depth (precipitable water vapor). Daytime (nighttime) measurements are indicated by a red (blue)

symbol. (a) 340 nm AOD, (b) 380 nm AOD, (c) 500 nm AOD, (d) 675 nm AOD, (e) 870

nm AOD, (f) 940 nm PWV, (g) 1020 nm AOD, (h) 1627 nm AOD.

Appendix A

Table A1 Coefficients for smoothing at the ROLO 32 wavelength.

Fig. A1 Coefficients for smoothing at the ROLO 32 wavelength.

Fig. A2 ROLO smoothed and adjusted reflectance.

Table 1 Examples of calibration coefficient $V_{S0}$ for the solar measurement.

| Wavelength (nm) | 340 | 380 | 400 | 500 | 675 | 870 | 940 | 1020 |
|---|---|---|---|---|---|---|---|---|
| $V_{S0}$ (×10⁻⁴) (A) | 0.1799 | 0.1882 | 1.603 | 3.174 | 3.444 | 2.299 | 1.055 | 1.077 |

| Wavelength (nm) | 1225 | 1627 | 2200 |
|---|---|---|---|
| $V_{S0}$ (×10⁻⁴) (A) | 0.9305 | 1.321 | 0.7873 |

Table 2 Measurement range before (current) and after modification (new) of POM-02. $I_n$ and $I_{n-1}$ are the upper and lower limits of the current (unit: A), respectively.

| Range no. | | Current | | | New | | |
|---|---|---|---|---|---|---|---|
| 1 | $I_1 - I_2$ | $2.5 \times 10^{-3}$ | – | $2.5 \times 10^{-4}$ | $2.5 \times 10^{-3}$ | – | $1.25 \times 10^{-4}$ |
| 2 | $I_2 - I_3$ | $2.5 \times 10^{-4}$ | – | $2.5 \times 10^{-5}$ | $1.25 \times 10^{-4}$ | – | $6.25 \times 10^{-6}$ |
| 3 | $I_3 - I_4$ | $2.5 \times 10^{-5}$ | – | $2.5 \times 10^{-6}$ | $6.25 \times 10^{-6}$ | – | $3.125 \times 10^{-7}$ |
| 4 | $I_4 - I_5$ | $2.5 \times 10^{-6}$ | – | $2.5 \times 10^{-7}$ | $3.125 \times 10^{-7}$ | – | $1.5625 \times 10^{-8}$ |
| 5 | $I_5 - I_6$ | $2.5 \times 10^{-7}$ | – | $2.5 \times 10^{-8}$ | $1.5625 \times 10^{-8}$ | – | $7.8125 \times 10^{-10}$ |
| 6 | $I_6 - I_7$ | $2.5 \times 10^{-8}$ | – | $2.5 \times 10^{-9}$ | $7.8125 \times 10^{-10}$ | – | $3.90625 \times 10^{-11}$ |
| 7 | $I_7$ | $2.5 \times 10^{-9}$ | – | 0.0 | $3.90625 \times 10^{-11}$ | – | 0.0 |
| | $I_n$ | $I_n = I_{n-1}/10$ | | | $I_n = I_{n-1}/20$ | | |

Table 3 Coefficients of the regression equation for reflectance correction factor C.

| Wavelength (nm) | $A_c$ | $B_c$ | RMS | $F_c$ | $A_c'=A_c/F_c$ | $B_c'=B_c/F_c$ | RMS/$F_c$ | No. of data |
|---|---|---|---|---|---|---|---|---|
| 340 | $1.3404\times10^{-5}$ | 0.98027 | 0.0152 | 0.8993 | $1.4905\times10^{-5}$ | 1.09010 | 0.0169 | 15 |
| 380 | $1.3512\times10^{-5}$ | 1.0674 | 0.0080 | 1.0153 | $1.3309\times10^{-5}$ | 1.05140 | 0.0079 | 15 |
| 400 | $3.0760\times10^{-6}$ | 1.0058 | 0.0055 | 0.95270 | $3.2287\times10^{-6}$ | 1.05570 | 0.0058 | 15 |
| 500 | $2.2487\times10^{-6}$ | 1.1600 | 0.0058 | 1.0184 | $2.2081\times10^{-6}$ | 1.13910 | 0.0057 | 15 |
| 675 | $4.8644\times10^{-6}$ | 1.0840 | 0.0048 | 0.95705 | $5.0827\times10^{-6}$ | 1.13260 | 0.0050 | 15 |
| 870 | $3.4967\times10^{-6}$ | 1.0855 | 0.0026 | 0.95705 | $3.6537\times10^{-6}$ | 1.13420 | 0.0027 | 15 |
| 940 | $7.2405\times10^{-8}$ | 1.1532 | 0.0404 | 1.0292 | $7.0352\times10^{-8}$ | 1.12050 | 0.0392 | 13 |
| 1020 | $6.7912\times10^{-6}$ | 1.0559 | 0.0078 | 0.97065 | $6.9966\times10^{-6}$ | 1.08790 | 0.0081 | 15 |
| 1225 | $9.0288\times10^{-5}$ | 1.0572 | 0.0328 | 1.0203 | $8.8491\times10^{-5}$ | 1.03620 | 0.0322 | 13 |
| 1627 | $2.3828\times10^{-5}$ | 1.0810 | 0.0237 | 1.0463 | $2.2774\times10^{-5}$ | 1.03310 | 0.0227 | 13 |
| 2200 | $3.7545\times10^{-6}$ | 0.95311 | 0.0386 | 0.97493 | $3.8511\times10^{-5}$ | 0.97763 | 0.0396 | 13 |

$C = A_c \cdot g^2 + B_c$

$g$: phase angle (degrees)

$F_c$: smoothing factor

[revised manuscript text omitted]

$C_1$ and $C_2$: coefficients of the regression line ($\tau_{Moon} = C_1 \cdot \tau_{Sun} + C_2$, $PWV_{Moon} = C_1 \cdot PWV_{Sun} + C_2$).

Table 10 Same as Table 4 except for the AERONET and POM-02 aerosol optical depth (precipitable water vapor).

| POM-02 | Wavelength (nm) | Bias | RMSE | CR | $C_1$ | C. I. of $C_1$ (95%) | $C_2$ | C. I. of $C_2$ (95%) | RMSE of reg. | No. of obs. |
|---|---|---|---|---|---|---|---|---|---|---|
| Sun | 340 | 0.0082 | 0.0091 | 0.9855 | 0.9722 | 0.0212 | 0.0086 | 0.0006 | 0.0040 | 242 |
| Moon | 340 | ----- | ----- | ----- | ----- | ----- | ----- | ----- | ----- | 0 |
| Sun | 380 | 0.0148 | 0.0153 | 0.9819 | 0.9840 | 0.0237 | 0.0151 | 0.0006 | 0.0037 | 249 |
| Moon | 380 | ----- | ----- | ----- | ----- | ----- | ----- | ----- | ----- | 0 |
| Sun | 500 | −0.0046 | 0.0050 | 0.9884 | 1.0060 | 0.0196 | −0.0046 | 0.0004 | 0.0021 | 243 |
| Moon | 500 | −0.0010 | 0.0054 | 0.7366 | 0.6369 | 0.1524 | 0.0044 | 0.0025 | 0.0045 | 59 |
| Sun | 675 | 0.0083 | 0.0085 | 0.9839 | 1.0279 | 0.0235 | 0.0081 | 0.0003 | 0.0018 | 245 |
| Moon | 675 | 0.0109 | 0.0112 | 0.8466 | 0.8119 | 0.1368 | 0.0123 | 0.0012 | 0.0024 | 56 |
| Sun | 870 | −0.0015 | 0.0020 | 0.9877 | 1.0459 | 0.0210 | −0.0018 | 0.0002 | 0.0013 | 241 |
| Moon | 870 | −0.0044 | 0.0059 | 0.7343 | 0.5275 | 0.1283 | 0.0004 | 0.0015 | 0.0028 | 58 |
| Sun | 940 | 0.0177 | 0.0223 | 0.9996 | 1.0712 | 0.0038 | −0.0025 | 0.0013 | 0.0054 | 259 |
| Moon | 940 | 0.0445 | 0.0535 | 0.9991 | 1.1610 | 0.0126 | 0.0035 | 0.0039 | 0.0086 | 59 |
| Sun | 1020 | 0.0017 | 0.0023 | 0.9796 | 1.0393 | 0.0269 | 0.0015 | 0.0002 | 0.0015 | 244 |
| Moon | 1020 | −0.0042 | 0.0080 | 0.4128 | 0.2846 | 0.1652 | 0.0038 | 0.0022 | 0.0045 | 58 |
| Sun | 1627 | 0.0020 | 0.0029 | 0.9828 | 1.1100 | 0.0359 | 0.0017 | 0.0003 | 0.0019 | 132 |
| Moon | 1627 | ----- | ----- | ----- | ----- | ----- | ----- | ----- | ----- | 0 |

Table A1 Coefficients for smoothing at the ROLO 32 wavelength

| Wavelength (nm) | Correction factor | Wavelength (nm) | Correction factor |
|---|---|---|---|
| 350.0 | 1.02766 | 763.7 | 1.00312 |
| 355.1 | 1.09314 | 774.8 | 0.95628 |
| 405.0 | 0.93705 | 865.3 | 0.94167 |
| 412.3 | 0.95166 | 872.6 | 0.96555 |
| 414.4 | 1.02732 | 882.0 | 0.94490 |
| 441.6 | 1.01667 | 928.4 | 0.97167 |
| 465.8 | 1.04970 | 939.3 | 1.04085 |
| 475.0 | 1.01461 | 942.1 | 0.99417 |
| 486.9 | 1.01748 | 1059.5 | 0.95872 |
| 544.0 | 1.02132 | 1243.2 | 1.02708 |
| 549.1 | 0.99098 | 1538.7 | 1.02616 |
| 553.8 | 1.02041 | 1633.6 | 1.04781 |
| 665.1 | 0.93882 | 1981.5 | 1.05865 |
| 693.1 | 0.99039 | 2126.3 | 1.08338 |
| 703.6 | 1.00576 | 2250.9 | 0.90003 |
| 745.3 | 0.99651 | 2383.6 | 0.98073 |

[Figure]

Fig. 1 Examples of sensor output for solar direct irradiances and scattered sky radiances from POM-02.

[Figure]

Fig. 2 An example of the measurements taken on Oct. 14, 2017 at NOAA/MLO. The phase angle of the moon (right y-axis) is from 117.6 to 118.0 degrees.

[Figure]

Fig. 3 Examples of the Langley plot in the visible and near-infrared region on Nov. 5, 2017. The y-axis is the equation in parentheses on the left-hand side of eq. (10).

(a) 340, 380, 400, 500 nm; (b) 675, 870, 1020 nm; (c) 940 nm, modified Langley method.

[Figure]

Fig. 4 Examples of the Langley plots in the shortwave infrared region.

[Figure]

Fig. 5 to be continued.

[Figure]

Fig. 5 Relationship between phase angle and reflectance correction factor $C = V_{m\theta}/V_{s\theta}$ in visible and near-infrared region. A regression curve ($C = A_c \cdot g^2 + B_c$, $g$ : phase angle) was also plotted.

to be continued.

[Figure]

[Figure]

Fig. 6 Relationship between phase angle and reflectance correction factor $C = V_{mb}/V_{sb}$ in shortwave infrared region. A regression curve ($C = A_c \cdot g^2 + B_c$, $g$ : Phase angle) was also plotted.

[Figure]

Fig. 7 Scatter plot of NIES/HSRL and POM-02 aerosol optical depth at 532nm. (a) Daytime (red), (b) nighttime (blue), (c) overlapping daytime (red) with nighttime (blue).

[Figure]

Fig. 8 Examples of time series of NIES/HSRL (red), POM-02 daytime (green), and nighttime (blue) aerosol optical depths at 532 nm. The phase angles (g) during the measurement periods were (a) g = −21.863 to 35.881 degrees, (b) g = 47.454 to 83.190 degrees, and (c) g = −19.150 to 21.573 degrees.

[Figure]

Fig. 9 Scatter plot of radiosonde and POM-02 precipitable water vapor. Daytime (nighttime) measurements are indicated by a red (blue) symbol.

[Figure]

Fig. 10 Same as Fig. 9 except for corrected POM-02 precipitable water vapor.

[Figure]

Fig. 11 Scatter plot of GPS and radiosonde precipitable water vapor.

[Figure]

Fig. 12 Scatter plot of PWV from GPS and corrected PWV from POM-02. (a) Daytime (red), (b) nighttime (blue), (c) overlapping daytime (red) with nighttime (blue).

[Figure]

Fig. 13 Examples of time series of GPS (red), POM-02 daytime (green), and nighttime (blue) corrected precipitable water vapor. The phase angles ($g$) during the measurement periods were (a) $g = -21.863$ to $35.881$ degrees, (b) $g = -19.150$ to $21.573$ degrees, and (c) $g = -55.145$ to $30.611$ degrees.

[Figure]

Fig. 14 to be continued.

[Figure]

[Figure]

Fig. 14 Scatter plot of the aerosol optical depth (precipitable water vapor) from the sun and the moon. (a) 340 nm AOD, (b) 380 nm AOD, (c) 400 nm AOD, (d) 500 nm AOD, (e) 675 nm AOD, (f) 870 nm AOD, (g) 940nm PWV, (h) 1020 nm AOD.

[Figure]

Fig. 15 to be continued.

**(g) 1020 nm**

[Figure]

**(h) 1627 nm**

[Figure]

Fig. 15 Scatter plot of AERONET and POM-02 aerosol optical depth (precipitable water vapor). Daytime (nighttime) measurements are indicated by a red (blue) symbol. (a) 340 nm AOD, (b) 380 nm AOD, (c) 500 nm AOD, (d) 675 nm AOD, (e) 870 nm AOD, (f) 940nm PWV, (g)1020nm AOD, (h) 1627 nm AOD.

[Figure]

Fig. A1 ROLO smoothed and adjusted reflectance.

[Figure]

Fig. A2 Coefficients for smoothing at the ROLO 32 wavelength.

Supplement

The time series of the AOD at 500 nm and PWV at Tsukuba for 5 months are shown in Fig. S1 and Fig. S2, respectively. These are non-cloud screened data.
In addition, the time series of the comparison between HSRL and POM-02 AOD and the time series of the comparison between GPS and POM-02 PWV are shown in Fig. S3 and Fig. S4.

Figure captions

Fig. S1 Time series of the AOD at 500 nm at Tsukuba for 5 months. (a) January, (b) February, (c) March, (d) April, (e) May.

Fig. S2 Time series of the PWV at Tsukuba for 5 months. (a) January, (b) February, (c) March, (d) April, (e) May.

Fig. S3 Time series of the comparison between HSRL and POM-02 AOD for 5 months. Red symbols are HSRL AOD, blue symbols are AOD in the nighttime, and green symbols are AOD in the daytime. The data are 15-minute averages.

Fig. S4 Time series of the comparison between GPS and POM-02 PWV for 5 months. Red symbols are GPS PWV, blue symbols are PWV in the nighttime, and green symbols are PWV in the daytime. The data are 30-minute averages.

[Figure]

Fig. S1 Time series of the AOD at 500 nm at Tsukuba for 5 months.
(a) January, (b) February, (c) March, (d) April, (e) May.

[Figure]

Fig. S2 Time series of the PWV at Tsukuba for 5 months.
(a) January, (b) February, (c) March, (d) April, (e) May.

[Figure]

Fig. S3 Time series of the comparison between HSRL and POM-02 AOD. Red symbols are HSRL AOD, blue symbols are AOD in the nighttime, and green symbols are AOD in the daytime. The data are 15-minute averages.

[Figure]

Fig. S4 Time series of the comparison between GPS and POM-02 PWV. Red symbols are GPS PWV, blue symbols are PWV in the nighttime, and green symbols are PWV in the daytime. The data are 30-minutes averages.

---

## Author Comment (AC6) · 16 Oct 2019

We also send main text as pdf file as supplement.

Please also note the supplement to this comment:
https://www.atmos-meas-tech-discuss.net/amt-2019-230/amt-2019-230-AC6-supplement.pdf

---

## Author Response (AR1)

Reply to comments

We would like to thank you for reading our manuscript and commenting on it.
The comments are copied and shown below in italic.

*Comment.*
*Anonymous Referee #1*

*Nocturnal aerosol optical depth measurements with modified skyradiometer POM-02*
*using the moon as a light source*
*by Akihiro Uchiyama et al.*

*General comments:*
*I have doubts on using AOD obtained from LIDAR as the truth. Usually is the opposite,*
*you use AOD from photometer to constrain the LIDAR results. This is also in some way*
*recognized by you at L474-477.*
=>

We think that the comparison requires data measured in a way independent of the
skyradiometer measurement. The HSRL is one of the instruments that can determine
the vertical distribution of the aerosol extinction coefficient. Although data processing of
HSRL is done under some assumptions, we used HSRL data.

*I suggest to read and eventually cite also the last paper from Barreto et al. 2019 about*
*intercomparison between CIMEL and PFR lunar versions.*
*https://doi.org/10.1016/j.atmosenv.2019.01.006*
=>

We cited a paper written by Barretto et al. (2019).
And, we added some sentences (see the text).

We added the following sentences in Introduction.
"In addition, a lunar photometer—the Moon Precision Filter Radiometer, LunarPFR
(Kouremeti et al. 2016)—has been developed by the Physical Meteorological Observatory
in Davos (PMOD), which serves as the World Radiation Center (WRC), based on the sun-
PFR experience. Using these instruments and stellar photometers, a multi-instrument
nocturnal intercomparison campaign was conducted to evaluate nighttime aerosol
measurements and lunar irradiance models (Barreto et al. 2019)."

*I think sections 2.2 and 2.3 could be merged. There are repetitions between them. I don't understand the last part of the last sentence of section 2.2: "depending on the aerosol optical depth....".*

=>

We merged section 2.2 and 2.3.

*Did you wrote an implementation of the ROLO equation? Did you used filter functions of just central wavelengths?*

=>

We used central wavelengths.

We mentioned it in the text.

*Eq4 In my opinion is not necessary to define C1 and C2, it's confusing to me.*

*Eq7 Same as for Eq4. By the way, you are using the same symbols C1 and C2*

=>

Coefficients C1 and C2 were introduced to clarify that eq. (4) is a linear function of airmass.

We rewrote equations (4), (7), and (10) and used different symbols.

*Finally, I suggest to ask a mother-tongue people to check the manuscript.*

=>

Whenever we submit a paper, our manuscript is checked by a person whose native language is English.

*Specific comments:*

*L34 I suggest "reflectance estimated by the Robotic..."*

=>

We rewrote.

*L34 Maybe a reference to the ROLO paper is required here*

=>

Kieffer and Stone (2005) is an important paper. However, we usually do not quote references in the abstract, so we do not quote it here.

L38 "visible and near-infrared"

=>

We rewrote.

L38 "This indicates..." or "could indicate"

=>

We rewrote.

L65 AERONET

=>

We rewrote.

L131 Remove "Prede Co Ltd., Japan". Is a repetition

=>

We removed it.

L157 Could you please explain this sentence? Where you get 2x10-5?

=>

We rewrote the sentence as follows.
"From Table 1, the calibration constants at 340 and 380 nm are $1.8\times10^{-5}$ and $1.9\times10^{-5}$ (about $2\times10^{-5}$), respectively. Therefore, the output for the direct lunar irradiance during the half moon is about $2\times10^{-5} \times 10^{-6} = 2\times10^{-11}$ in the 340 and 380 nm channels."

L177 "Therefore, it is difficult..."

=>

We rewrote.

L220 The equation is reported without any introduction words.

=>

We inserted the following sentence.

   "The empirically derived analytic form based on the primary geometric variables is as follows: "

L305-308 Repetition. Please remove

=>

 We removed these sentences.

*L373-374 Even if the meaning is clear to me, I suggest to rephrase the sentence in a clearer way.*

=>

We rephrased the text.

Please see the text.

*L394-395 Remove the repetition of "the detector output" -> "and hence may be..."*

=>

We removed "the detector output".

*L406-409 Please rephrase the sentence in a clearer way.*

=>

We rephrased the text.

Please see the text.

*L413 The sentence about the calibration factors at 940 nm is not clear to me.*

=>

We rephrase the sentence.

"The absorption band of water vapor is at the 940 nm wavelength. Water vapor in the atmosphere tends to fluctuate. Therefore, it is difficult to make accurate Langley plots, and the accuracy of both $V_{S0}$ and $V_{m0}$ is poor."

*L418 Please check the "C(="*

=>

"C(=" is correct.

*L448-451 This sentence is a repetition.*

=>

We removed the second sentence.

*L491 UTC, not UCT*

=>

We replace UCT with UTC.

*L494-496 You repeated many times in the text this explanation about the statistics shown in the tables. I think one time is enough.*

*=>*

We changed the expression after the second.

*L602-603 The sentence about the ROLO model is not clear to me.*

*=>*

We rephrase the text.

*Please rephrase Table1 Which are the units? Amperes?*

*=>*

We inserted the unit.

*Fig2, Fig3 Maybe you could improve/simplify the labels of the y axes*

*=>*

In the caption of Figs. 2 and 3 (new Figs. 3 and 4), we added the explanation that " The y-axis is the equation in parentheses on the left-hand side of eq. (11).".

Reply to comments

We would like to thank you for reading our manuscript and commenting on it.
The comments are copied and shown below in italic.

*Comment.*
*Anonymous Referee #2*

*Nocturnal aerosol optical depth measurements with modified skyradiometer POM-02*
*using the moon as a light source*
*by Akihiro Uchiyama et al.*

*General comments:*
*Overall, this is a well-written paper describing the modification of the current PREDE-POM radiometers to extend the aerosol monitoring at night-time using the moon as the light source. Considering this to be the reference instrument of one of the most important ground-based networks such as Skynet, the adoption of this new system will provide valuable information for atmospheric research and will certainly be widely used. I consider that this manuscript fits perfectly into the scope of AMT. I recommend publishing the manuscript, but there are some important discussion points and details that I would like the authors to address before its final publication:*

*· This referee is concerned about the lack of continuous lunar measurements shown in this paper. The authors measured 2 moon cycles in MLO but they do not show any results in terms of AOD. Furthermore, they have 6 moon cycles in Tsukuba, but only a few days are shown in figure 7. They claim they can perform measurements at higher phase angles, but there is no evidence of this in the text. Figure 7 offers no information as to what phase angle corresponds with these measurements, and they do not constitute proof of continuous lunar measurements in themselves (see also specific comments).*
=>
In Tsukuba, clear days with few clouds do not last long; at best only 3 or 4 days. Even at MLO, in the late morning and afternoon hours, marine aerosol and clouds reaches the observatory during the marine inversion boundary layer breakdown under solar heating. Therefore, we selected the continuous days with few clouds and presented the data on these days as measurement examples.
The time series of AOD at 500 nm and PWV in Tsukuba for 5 months is shown in

Supplement. These are non-cloud screening data. In addition, the time series of comparison between HSRL and POM-02 AOD for 5 months and the time series of comparison between GPS and POM-02 PWV for 5 months are also shown in Supplement. In Fig.7 (new Fig. 8) and Fig.12 (new Fig. 13), the phase angles during the measurement periods were added to the captions.

*· Moreover, the choice of the NIES/HSRL as the only validation analysis of this paper seems to be an important weak point (it needs interpolation or the assumption of a constant extinction coefficient at altitudes of less than 500m). There is a Cimel CE318-T taking measurements at MLO during the period the Prede was calibrating there. I suggest the authors include this interesting comparison in their study. Furthermore, the statistics involved in the day/night/day continuity of AOD will give important information about the instrument's performance.*
=>

We think that the comparison requires data measured in a way independent of the skyradiometer measurement. The HSRL is one of the instruments that can determine the vertical distribution of the aerosol extinction coefficient. Although data processing of HSRL is done under some assumptions, we used HSRL data.

We compared AODs (PWVs) before and after the sunrise and the sunset.
Examining the continuity of AOD (PWV) is an effective means for observation site where the atmosphere remains stable. In Tsukuba, the stable condition does not last for a long time. The comparison was made by selecting data with small variations.

We compared AERONET and POM-02 data.
In the nighttime, the atmosphere observed at MLO was pristine, and most of the AOD at 500, 675, 870, and 1020 nm were below 0.02. Considering that the accuracy of the calibration constant is 0.5 to 1%, it is difficult to compare the AOD of AERONET and POM-02.
The AERONET data used here is "level 2.0" in the daytime and "level 1.5" in the nighttime. Because there is no level 2.0 nighttime data. AERONET "level 1.5" is cloud-screened data but may not have final calibration applied.  These data are not quality assured. AERONET "level 2.0" is pre- and post-field calibration applied, cloud-screened, and quality-assured data.
The primary purpose of MLO measurements is to acquire data for Langley calibration.

· *In the manuscript there is sometimes a lack of required scientific and analytical rigor and therefore, I suggest that potential subjective sentences be avoided. I would like to highlight the following examples:*

- *The ability of POM-02 to perform measurements beyond the quarters. In the conclusions the authors stated that this instrument is able to measure up to 120 degrees in phase angle. This value is quite surprising because it is the first time I have read it in the text. Notwithstanding, considering this to be the limit of your instrument's measuring capabilities, the authors must provide proof of that. Sections 2 and 3 describe a sensor able to perform measurements between +/-90 degrees and the authors claim that it is possible to extend this range if the instrument is accurately installed. Firstly, this assumption is vague and imprecise, and secondly there is no evidence in the manuscript showing this instrument is capable of providing measurements in this claimed phase angle range.*

=>

The measurement example on Oct. 14, 2017 at MLO was shown. In this example, the phase angle is from 117.6 to 118.0 degrees.

Also, we rewrote Sections 2.2 and 2.3.

- *The authors use only the ROLO model in terms of reflectance, assuming the relative change in the model's reflectance is correct. But as Kieffer and Stone (2005) stated, direct dependence on solar model will cancel itself out as long as the same model is used in going from irradiance to reflectance and back again. That means, the authors are introducing an "uncounted" error in this step, and they are not correcting their model implementation by using the Apollo spectra either. The authors should give some discussion about this effect, taking into account that they attribute the residuals of the C coefficient solely to errors of the ROLO model itself or to lunar librations. There are also other sources of error such as interpolation/extrapolation of the ROLO coefficients, temperature correction, Langley fitting, possible nonlinearity of PREDE-POM sensor, noise, among others.*

=>

As you said, Kieffer & Stone (2005) recommends using the same solar model (Wehrli, 1985) as Kieffer & Stone (2005) when converting ROLO reflectance to irradiance. This is because the Wehrli solar model was used to convert the original ROLO irradiance data to reflectance for developing the disk reflectance model. As you known, the value of the solar spectrum is different from model to model.

We don't use irradiance here. We use only reflectance data.

The coefficient C used here includes a coefficient for smoothing based on Apollo 16 Samples. Kieffer & Stone (2005) does not show the values of coefficients a and b of Aapollo = (a + bλ) Arolo (7,7,0,0), where Aapollo is reflectance based on Apollo 16 samples, Arolo(7,7,0,0) is ROLO reflectance for specific geometric configuration $(g = 7\deg, \Phi = 7\deg, \theta = 0, \phi = 0)$. These coefficients are necessary for smoothing. If we determine coefficients a and b by ourselves, there is a possibility that a new error will occur. Therefore, the original ROLO reflectance (Arolo) was used instead of the smoothed Arolo. However, the values of coefficients a and b that we determined are shown in Appendix A. The values interpolated into the wavelength of POM-02 are also shown in Table 3. To determine coefficients a and b, Apollo samples reflectance data is necessary. The values of coefficients a and b are dependent on the interpolation method of reflectance table and the accuracy of the value read from the figure.

As you pointed out, the residuals of C come from many factors.
We rewrote the text.

- *Similar precision and accuracy of night-time measurements in comparison to daytime: I suggest the authors present some evidence which justifies this statement or avoid giving vague assumptions. I understand that an uncertainty analysis could be out of the scope of this paper (although it would be very illuminating). However, without this study, the authors should not state that the uncertainty at daytime is similar to the one at night-time.*

=>
We want to know if there is a difference between daytime and nighttime measurements. We compared the measurements for the daytime and nighttime with measurement data which was recorded independently of POM-02 and has the same accuracy and precision in the daytime and nighttime.

The accuracy of the nighttime calibration constant is lower than that for the daytime. The measurement S/N in the nighttime is also worse than that in daytime. Considering these facts, even if there is no statistically significant difference, the magnitude of the error in the AOD (PWV) during the nighttime is not always the same as during the daytime. Further research and development are required.

*Specific comments:*

*This is not the first time that an attempt for ROLO correction has been published. Therefore, I recommend including some discussion about similar corrections already presented in previous publications.*

=>

We cited references.

*Amplification in the new PREDE-POM: I am not sure by reading the text if the new instrument includes amplification or not. I read on page 3, lines 170-172, that it can use amplifiers in the visible and nIR spectral range (1 to 7). However, in Eq. 2, there is no amplification used between sun and lunar measurements. Could you please clarify this?*

=>

The same amplifier was used for both the solar and lunar measurements.

The sensor output takes into account the magnification of the amplifier

We rewrote Section 2.

*This is a technical paper aiming to extend AOD capabilities to night-time. Nonetheless, the authors presented a correction of the lunar irradiance model without showing its impact in terms of AOD. Furthermore, this goal seems very ambitious, taking into account that the continuous operation of this instrument has not been appropriately verified.*

=>

The purpose of this paper is to show that POM-02 can measure AOD and PWV using the moon as the light source. If the ROLO reflectance is not corrected, the calibration constant has an error depending on the phase angle. If not corrected, there will be an error in the optical thickness depending on the phase angle.

POM-02 has been shown to be capable of continuous measurement during daytime measurements. The current POM-02 has the ability to measure the direct irradiance from the moon for some channels in the visible and near-infrared wavelength region without requiring modification. In this paper, only the function to track the moon has been added. We believe that POM-02 can measure continuously even during the nighttime as well as during the daytime.

*Page 13, lines 404-417: How many points do you use (C coefficients) to perform your ROLO correction? What about the residuals? Do you use the residuals to estimate the precision of this correction by 1%? Please clarify.*
=>
The number of data used to determine the coefficient of C and the residual are added to Table 3. The residual value is the error of the calibration constant.

*Page 15, lines 469-472: there are references to "similar time variations" or "systematic differences" without quantifying them. The authors also claimed that this figure constitutes a continuous series, but it contains only 11 different nights from 3 different moon cycles (with 4, 4 and 3 nights, respectively). You do not provide any more data for an entire moon cycle, and is this because of clouds, technical problems… Please clarify.*
=>
See reply to general comment.
In Tsukuba, it is unlikely that a clear day during the entire moon cycle lasts. At best, a clear day with no clouds lasts only for 3 or 4 days. In these figures, clear days with few clouds during the 3 or 4 days and with a long measurement time of the moon were selected. We added the word "qualitatively" to the text.

*I feel lost with units in Section 2.1. You write Amps almost everywhere, but sometimes I think you are talking about physical units (irradiance or radiance units). Figure 1 presents solar direct and scattered irradiance in Amps. Why do you convert everything into current intensity?*
=>
We have never converted the measurement values into irradiances. The raw sensor output value of POM-02 is current (unit: A). The transmittance is necessary for the estimation of the AOD. To obtain transmittance, a calibration constant is required; the instrument's output for the sun or moon outside the atmosphere.

*Technical comments:*
*Page 1, line 33: Please define MLO.*
=>
We rewrote.

*Page 1, line 33: I read in the text (page 8 line 258) that the calibration period is September-November. Please verify.*

=>

*We rewrote.*

*Page 2, line 44: Please define MRI/JMA.*

=>

*We rewrote.*

*Page 2, line 44: Please define NIES.*

=>

*We rewrote.*

*Page 2, line 50: As written in general comments, the authors should avoid making vague assumptions.*

=>

*See above.*

*Page 2, line 65: there is a typo in AERONET.*

=>

*We rewrote.*

*Page 3, lines 79-84: Please homogenize if you write acronyms or not.*

=>

*We rewrote.*

*Page 3, line 98: "in many cases" Please be more specific. Are the authors talking about elastic lidars?*

=>

*We rewrote this paragraph.*

*Page 5, section 2.2: I feel the FOV of the instrument is missing here. I assume it is the same as the non-modified Prede, but I consider it will be useful for the reader to include this information.*

=>

*We added the sentences in Section 2.*

*In the modification of the POM-02 for solar observation, only the amplifier and the position sensor were changed. The other components, e.g., detectors, filters, and lenses,*

are not changed. Therefore, the magnitude of the solid view angle (field of view) for the new POM-02 is the same as in the non-modified POM-02.

*Page 8, line 257: How many measurements?*
=>
We wrote the number of days and nights used to determine the calibration constants.

*Page 9, section 5.1: I recommend the authors to separate this section into two, one for daytime calculations and another one for the methodology applied to night-time. I consider it will certainly improve the readability of this section.*
=>
We divided section 5.1 into two sections.

*Page 9, line 370: Is the Rm distance also expressed in AU? ROLO refers the reference to the mean Earth-Moon distance in km (384400km).*
=>
As you pointed out, it is a mistake.

*Page 10, line 325: Why the authors say "it is often used"? Because they use it sometimes and sometimes not? Please clarify.*
=>
We deleted "often".

*Page 10, line 351: Please be clearer about "several atmospheric models".*
=>
Please see a reference.

*Page 13, line 407: "…the moon and the sun".*
=>
We rewrote the text.

*Page 19, line 639: This is the first time I have read 120 degrees in the text.*
=>
We showed an example of measurement.

*Figure 2: Please label each figure and include some information about each sub-plot. According to page 12, line 392 and Eq. 10, the y-axis of this plot must be*

$$\ln(\pi \cdot V \cdot R_S^2 \cdot R_m^2 / A \cdot \Omega) \text{ ?}$$

=>

The y-axis is the equation in parentheses on the left-hand side of eq. (11).

*Figures 6 and 7: Please include information about wavelength.*

=>

We add wavelength.

Reply to comments

We would like to thank you for reading our manuscript and commenting on it.
The comments are copied and shown below in italic.

*Comment.*
*Thomas Stone*

*Nocturnal aerosol optical depth measurements with modified skyradiometer POM-02*
*using the moon as a light source*
*by Akihiro Uchiyama et al.*

*The article "Nocturnal aerosol optical depth measurements with modified skyradiometer*
*POM-02 using the moon as a light source" presents aerosol optical depth (AOD)*
*measurements acquired at night using the Prede POM-02 radiometer. The exo-*
*atmospheric lunar irradiances used in this work for AOD retrievals are obtained from*
*an implementation of the USGS ROLO lunar model [Kieffer and Stone, 2005 (hereafter*
*K&S)]. The approach presented in this paper utilizes the direct outputs of the ROLO*
*lunar disk reflectance model, K&S Eq. 10, rather than using the model to generate lunar*
*irradiance as it is intended. These direct outputs of K&S Eq. 10 are then interpolated to*
*the wavelengths of the POM-02 instrument spectral bands. There are significant*
*problems with this approach, and it is not recommended to use the ROLO model in this*
*way.*

*The lunar disk reflectance spectrum produced from computing K&S Eq. 10 exhibits a*
*large amount of spectral structure, as can be seen in the example of Figure 1 of this*
*comment. This structure is the result of the development process for the ROLO model,*
*where the model coefficients were determined for each of the ROLO bands independently.*
*However, the actual reflectance spectrum of the Moon is known to be smooth, with only*
*broad, shallow absorption features [e.g. Pieters and Mustard, 1988; McCord et al., 1981].*
*To remove the artificial spectral structure produced by K&S Eq. 10, the USGS ROLO*
*system fits these outputs with a reference lunar reflectance spectrum. This reference*
*spectrum was developed from laboratory spectra of returned Apollo samples, and it is*
*considered representative of the lunar disk as a whole in terms of spectral content (not*
*absolute reflectance). The fitting process smoothes the results of Eq. 10 to give a realistic*
*lunar reflectance spectrum. Figure 1 shows an example of the outcome of this process. It*

*should be apparent from this figure that interpolating between the direct results for the ROLO bands (square symbols) generally will not produce values that align with the fitted spectrum (solid line).*

*The recommended usage of the ROLO model is to fit the reference reflectance spectrum to the outputs of Eq. 10, where these have been computed for the particular phase angle and librations of the instrument's Moon observation, then convolve this fitted spectrum with the sensor band spectral response and the solar spectral irradiance to give lunar irradiance, as:*

$$E_M = \frac{\Omega_M}{\pi} \frac{\int A_{fit}(\lambda) E_{Sun}(\lambda) S(\lambda) d\lambda}{\int S(\lambda) d\lambda} \tag{1}$$

*where $E_M$ is the lunar spectral irradiance, $\Omega_M$ is the solid angle of the Moon, $A_{fit}(\lambda)$ is the fitted lunar reflectance spectrum, $E_{Sun}(\lambda)$ is the solar spectral irradiance, and $S(\lambda)$ is the sensor band spectral response. The result of Equation 1 must then be corrected for the actual Sun-Moon and Moon-sensor distances of the instrument's Moon observation. The solar spectral irradiance used in this step should be the PMOD-WRC model published by C. Wehrli (ftp://ftp.pmodwrc.ch/pub/publications/pmod615.asc). This is because the Wehrli solar model was used to convert the original ROLO irradiance data to reflectance for developing the disk reflectance model, and the same solar model must be used to convert reflectance back to irradiance (Equation 1) to avoid errors caused by differences in the solar spectral models.*

==>

We do not use irradiance values. All we need is a reflectance value.

In our method, we used the original ROLO model (K & S Eq.10), which is not smoothed, for the following reasons.

(1) We do not use irradiance values.

(2) The smoothed ROLO reflectance spectrum is finally the original ROLO reflectance multiplied by a constant factor (see text).

(3) We replaced Arolo with CArolo, and determined C. This coefficient C includes the effects of smoothing (see the text).

(4) In K & S, the Apollo composed spectra Aapollo and Arolo are related by Aapollo = (a + bλ) Arolo. However, the values of the coefficients a and b are not shown.

(5) We can use the numerical values for the reflectance of Apollo 16 sample 62231

(http://www.planetaly.brown.edu/pds/AP62231.html). However, values of the reflectance of Apollo 16 sample 67455 must be read from Fig. 8 of Pieters & Mustard (1988).

Considering these things, we rewrote the text so that the relationship between the smoothed Arolo and the original Arolo can be understood.
We also showed the smoothing factor that we determined in Appendix.